# Noise-to-Process Transformation: A Weak-Prior Paradigm for Single-Trajectory Stochastic Process Modeling

## Abstract

Stochastic processes offer a principled framework for trajectory-level uncertainty modeling from limited observations. Prior-driven methods (e.g., Gaussian processes) remain viable with scarce data but hinge on strong structural priors, whereas data-driven meta-approaches learn flexible representations yet typically require multi-trajectory supervision. To achieve flexibility from a single trajectory without strong priors, we introduce a *noise-to-process (N2P)* paradigm: a shared base-noise process $Z$ is pushed through a single measurable generator $G_\theta$ to produce a full trajectory $X = G_\theta(Z)$, making projective consistency intrinsic by design. Instantiating the paradigm, we propose *Deconvolution-Based Process Transformation* (DBPT), a deconvolution-based generator that captures long-range, inter-temporal dependence. Across synthetic and diverse real single-trajectory tasks, DBPT delivers flexible uncertainty modeling and competitive performance to prior- and data-driven baselines.

## 1 Introduction

Uncertainty modeling aims to characterize the full predictive distribution of observations rather than point predictions. It's particularly valuable in regimes that often yield only a single noisy trajectory with few samples—conditions that make point predictions brittle and render data-hungry multi-trajectory or meta-learning approaches impractical. In high-fidelity CFD wing simulations Mohammad Zadeh et al. (2016), for instance, each configuration is expensive and typically yields a single, noisy trajectory corrupted by solver jitter, and turbulence-model uncertainty. In this setting, stochastic processes provide a principled framework for modeling trajectory-level latent dynamics and uncertainties, even when only a single noisy trajectory is available, enabling reliable probabilistic predictions under heterogeneous noise Ross (1995).

Existing approaches to stochastic-process modeling largely fall into two methodological paradigms: prior-driven and data-driven. Prior-driven approaches, e.g., Gaussian processes (GPs) MacKay et al. (1998) and SDE-based models Øksendal (2003), encode domain knowledge via structural priors. These methods are often applicable in single-trajectory settings and, when the prior is well specified, can deliver calibrated uncertainty estimates from limited data. Yet their flexibility is restricted to the chosen prior family, making performance sensitive to prior misspecification Wilson et al. (2016); Vaisband et al. (2025). To mitigate this limitation, recent studies couple neural networks with traditional priors to increase representational flexibility Sendera et al. (2021); Tzen & Raginsky (2019); Wilson et al. (2016). Nevertheless, because learning remains anchored to a predefined prior scaffold, generalization and robustness can still be limited Ober et al. (2021); Oh et al. (2025).

Data-driven paradigms, e.g., neural processes (NPs) Garnelo et al. (2018a), can learn flexible representations via amortized inference from multi-trajectory data, while requiring only weak prior assumptions. However, in the single-trajectory regime, these methods often suffer from amortization gaps Marino et al. (2018) and poorly calibrated uncertainty, thereby limiting extrapolative performance and reliability.

Taken together, these considerations motivate a central question: Can we design a weak-prior stochastic-process modeling paradigm that learns effectively from a single trajectory, preserving the applicability of prior-driven methods while retaining the flexibility of data-driven approaches?

To this end, we introduce a *noise-to-process (N2P)* paradigm for weak-prior stochastic process modeling in the single-trajectory setting. The central idea is to learn a parameterized, pathwise generator $G_\theta$ that maps a readily sampled, shared base-noise process $Z$ to an entire trajectory $X = G_\theta(Z)$ consistent with the observed data, which makes it possible to directly capture the correlations between adjacent variables. This *single-generator + shared-noise* structure makes finite-index marginals projections of the same joint sample, thereby rendering projective consistency intrinsic by design. Building on this paradigm, we develop *Deconvolution-Based Process Transformation* (DBPT), a deconvolution-based instantiation that captures inter-temporal dependence. Crucially, DBPT combines representational flexibility with reliable uncertainty quantification in few-shot, single-trajectory regimes, without relying on strong priors or multi-trajectory supervision. The main contributions are summarized as follows:

- We formalize a learnable, weak-prior noise-to-process representation: a shared base-noise process drives a single measurable generator that outputs an entire trajectory in one pass. This structure internalizes projective consistency and is compatible with Kolmogorov extension on denser index sets. The design is once-for-all and index-agnostic, decoupling parameter count from index-set size.

- N2P is instantiated with a deconvolution-based process transformation (DBPT) which is designed to model long-range, inter-temporal dependencies from a single observed trajectory, while enabling uncertainty.

- On synthetic and real tasks, DBPT delivers flexible representations and calibrated uncertainty compared to prior- and data-driven baselines in the single-trajectory setting.

## 2 A Learnable Noise-to-Process Representation

We construct a *learnable, weak-prior* design principle for single-trajectory learning: a *shared base-noise process* driving a *single measurable generator* that produces an entire trajectory in one pass. This construction *ensures projective consistency by design*, decouples parameterization from index-set size, and preserves *cross-index dependencies*.

### 2.1 Learnable N2P Representation on a Discrete Grid

Let the index set be a finite or countable grid $\mathcal{T} = \{t_1, t_2, \ldots\}$ and the state space $\mathsf{S} = \mathbb{R}^{d_y}$ (standard Borel). Let $(\mathcal{Z}, \mathcal{B}(\mathcal{Z}))$ be a standard Borel space and let $Z = (Z_1, Z_2, \ldots) \sim \nu$ be an i.i.d. base-noise process with product law $\nu$ (e.g., $\mathcal{Z} = [0,1]$ with Lebesgue measure, or $\mathcal{Z} = \mathbb{R}$ with Gaussian factors; these choices are measurably isomorphic).

**Definition 1** (Learnable noise-to-process (N2P) representation). *A parameterized measurable generator $G_\theta : \mathcal{Z}^{\mathbb{N}} \to \mathsf{S}^{\mathcal{T}}$ defines a trajectory $X = G_\theta(Z) = (X_t)_{t \in \mathcal{T}}$. We call this a* learnable N2P *representation if the same $G_\theta$ and the same shared base-noise process $Z$ are used to produce* all *coordinates $\{X_t\}_{t \in \mathcal{T}}$ in one shot.*

This definition encodes our weak structural prior: *shared noise + single generator*. Parameters $\theta$ are learned from a single observed trajectory; uncertainty is induced by resampling $Z$. Evaluation maps and pushforwards are recalled in Appendix B, Lemmas 8 and 9.

**Proposition 2** (Well-defined process on $\mathcal{T}$). *Let $\mu_\theta \triangleq \nu \circ G_\theta^{-1}$ be the pushforward law on $\mathsf{S}^{\mathcal{T}}$. Then $X \sim \mu_\theta$ is a well-defined stochastic process on the grid $\mathcal{T}$. Full proof: see Proposition 10.*

**Proposition 3** (Intrinsic projective consistency). *For any finite $I \subset \mathcal{T}$, let $\pi_I : \mathsf{S}^{\mathcal{T}} \to \mathsf{S}^I$ be the coordinate projection and define $\mu_{\theta,I} = \pi_I \# \mu_\theta$. Then for all $J \subset I$,*

$$\pi_J^I \# \mu_{\theta,I} = \mu_{\theta,J}.$$

Sketch. *Let $\pi_J^I : \mathsf{S}^I \to \mathsf{S}^J$ be the coordinate projection and $\pi_J^{\mathcal{T}} : \mathsf{S}^{\mathcal{T}} \to \mathsf{S}^J$ the global projection; then $\pi_J^{\mathcal{T}} = \pi_J^I \circ \pi_I$, and functoriality of pushforwards gives $\pi_J^I \#(\pi_I \# \mu_\theta) = (\pi_J^I \circ \pi_I) \# \mu_\theta = \pi_J^{\mathcal{T}} \# \mu_\theta = \mu_{\theta,J}$. Full proof: see Appendix B, Proposition 11.*

**Remark 4.** *The novelty is a* learnable, weak-prior *structure that internalizes consistency: a shared base-noise process and a single measurable generator $G_\theta$ yield the full trajectory in one pass, making all finite-dimensional marginals projections of one joint sample, while directly modeling local dependencies.*

## 2.2 COMPATIBILITY WITH KOLMOGOROV EXTENSION

If one wishes to pass from a nested grid $\mathcal{T}_1 \subset \mathcal{T}_2 \subset \cdots$ (whose union is dense in a continuum index set) to a full process on the larger index set, Propositions 2–3 produce, for each $n$, a marginal law $\mu_\theta^{(n)}$ on $\mathsf{S}^{\mathcal{T}_n}$, and these laws are pairwise consistent by Proposition 3. *Therefore, by Kolmogorov's extension theorem*, there exists a process law on the larger index set having $\{\mu_\theta^{(n)}\}$ as its finite-grid restrictions. This is a *compatibility* statement; it requires no additional modeling assumptions and does not affect training, which operates on the discrete grid. A formal compatibility statement appears in Appendix B.2 (Corollary 13).

## 2.3 DECONVOLUTION-BASED PROCESS TRANSFORMATION

Building on the foregoing N2P representation, modeling a stochastic process by directly learning a pathwise generator $G_\theta$ that maps a shared i.i.d. *base-noise process* to a trajectory is theoretically sound. However, designing a transformation capable of capturing complex dependencies among coordinates remains challenging. To address this, we introduce *Deconvolution-Based Process Transformation*, a model that learns such a transformation end-to-end from single-trajectory supervision.

The flowchart of DBPT is shown in Fig. 1. DBPT, denoted $G_\theta$, comprises two components: a *Noise encoder* $h_{\theta_h}$ and a *Deconvolution-based process decoder* $g_{\theta_g}$. The *noise encoder* takes the base-noise process $Z$ as input and produces a preliminary feature $r$. The *deconvolution-based process decoder* then transforms $r$ into a target trajectory consistent with the observations. During training, the loss is applied at the observed indices $\tau_o$, guiding the transformation toward agreement with $O$.

DBPT Model

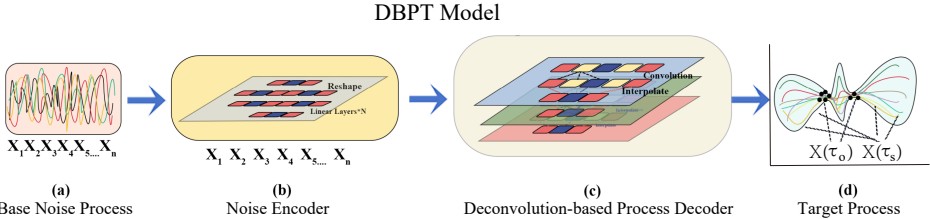

| (a) | (b) | (c) | (d) |
| --- | --- | --- | --- |
| Base Noise Process | Noise Encoder | Deconvolution-based Process Decoder | Target Process |

Figure 1: The flowchart of the DBPT consists of four sequential parts: (a) the input of the base noise process $Z$, (b) the noise encoder, (c) the deconvolution-based process decoder, which establishes the inter-temporal dependence, and (d) the output target stochastic process.

### 2.3.1 DBPT MODEL

**Discretized setup.** Let the discretized index set be $\mathcal{T} = \{t_1, \ldots, t_{\mathcal{T}}\}$ and the state space $\mathsf{S} = \mathbb{R}^{d_y}$. Let $\tau_o \subseteq \mathcal{T}$ be the observed indices and $\tau_s := \mathcal{T} \setminus \tau_o$ the unobserved ones. Let $O \in \mathbb{R}^{|\tau_o| \times d_y}$ denote the observations (ordered according to $\tau_o$).

We realize the shared base-noise process on the grid by drawing i.i.d. noise per index: for each $t \in \mathcal{T}$, sample $Z(t) \sim \mathcal{N}(0, I_{d_z})$ and write $Z(\mathcal{T}) = \{Z(t)\}_{t \in \mathcal{T}} \in \mathbb{R}^{|\mathcal{T}| \times d_z}$, with arbitrary $d_z$.

DBPT implements a pathwise generator via a composite map with parameter partition

$$G_\theta := g_{\theta_g} \circ h_{\theta_h}, \qquad \theta = (\theta_h, \theta_g),$$

$$\widehat{X}(\mathcal{T}) := G_\theta\big(Z(\mathcal{T})\big) \in \mathbb{R}^{|\mathcal{T}| \times d_y},$$

where $\theta_h$ and $\theta_g$ collect the trainable parameters of the noise encoder $h$ and the deconvolution-based process decoder $g$, respectively. We write $\widehat{X}$ for a sample trajectory produced by $G_\theta$ (i.e., a realization of $X = G_\theta(Z)$ on the grid $\mathcal{T}$).

**Noise encoder** Starting from the i.i.d. base-noise field $Z(\mathcal{T})$ (which supplies stochasticity while leaving dependencies to be learned downstream), the encoder maps $Z : \mathcal{T} \to \mathbb{R}^{d_z}$ to an intermediate representation $r \in \mathbb{R}^{|\mathcal{T}| \times c}$:

$$r = h_{\theta_h}\big(Z(\mathcal{T})\big),$$

where $h_{\theta_h}$ is a pointwise MLP.

**Deconvolution-based process decoder**    The preliminary features $r$ may exhibit long-range correlations, hierarchical patterns, non-stationarity, and nonlinear interactions. To transform $r$ into an output trajectory with coherent inter-temporal dependence, we employ a multi-layer deconvolution-based decoder that enforces cross-location consistency across scales:

$$\widehat{X}(\mathcal{T}) = g_{\theta_g}(r).$$

The feature maps $r$ are processed by a sequence of deconvolution layers, each consisting of up-sampling and convolution. Upsampling expands the spatial/temporal resolution by an integer factor, while the convolution with shared kernels couples neighboring positions, injecting spatial coherence across the grid. Stacking multiple deconvolution blocks yields a multi-scale receptive field that captures both local and long-range dependencies and accommodates non-stationarity via hierarchical refinement, while being flexible enough to fit complex dependencies from single-trajectory supervision Chen et al. (2022). In effect, this decoder propagates supervision from observed to unobserved indices through shared convolutions and multi-scale upsampling, yielding trajectories with coherent inter-temporal dependence and enabling arbitrary-form, flexible uncertainty quantification.

### 2.3.2 TRAINING

We train DBPT by minimizing a masked mean-squared error (MSE) on the observed subset $\tau_o$. Let $R_{\tau_o} : \mathbb{R}^{|\mathcal{T}| \times d_y} \to \mathbb{R}^{|\tau_o| \times d_y}$ denote the selection operator that extracts the rows indexed by $\tau_o$. With $\widehat{X}(\mathcal{T}) = G_\theta\big(Z(\mathcal{T})\big)$ and $O$ ordered consistently with $\tau_o$, the loss is

$$\mathcal{L}(\theta) = \mathbb{E}_Z \left[ \frac{1}{|\tau_o|} \left\| R_{\tau_o} \widehat{X}(\mathcal{T}) - O \right\|_F^2 \right],$$

where we approximate the expectation by Monte Carlo with fresh noise draws per iteration. Minimizing $\mathcal{L}$ enforces agreement with the observations on $\tau_o$. Although no direct supervision is available on $\tau_s$, the deconvolution-based process decoder $g_{\theta_g}$ propagates observational constraints through shared kernels and multi-scale upsampling, inducing coherent inter-temporal structure over $\tau_s$. After training, trajectory samples consistent with the observations are obtained by resampling $Z$ and evaluating $G_\theta\big(Z(\mathcal{T})\big)$; repeated draws provide uncertainty at unobserved locations, while finite-index marginals remain intrinsically projectively consistent by construction (Prop. 3).

**Theory pointers.**    Generalization and mean-calibration guarantees for the masked training protocol are provided in Appendix C. Identifiability at the path level and at the process-law level (up to noise isomorphism) is discussed in Appendix D.

## 3   RELATED WORK

In this section, we review related work on stochastic process modeling, covering both prior-driven and data-driven approaches. Given the conceptual proximity between our framework and modern generative models, we also include a brief discussion of generative models—although they are not stochastic-process methods per se.

**Prior-Driven Approaches.**    Prior-driven methods encode domain knowledge via explicit structural priors and are often applicable in single-trajectory settings. In GPs, a prior over functions is specified as $f \sim \mathcal{GP}(m, k_\theta)$, where kernels (e.g., RBF, Matérn) encode inductive biases for trajectory-level regression Seeger (2004). Flexibility of GP-based models is enhanced via output-space warping Lázaro-Gredilla (2012), layered GP compositions Damianou & Lawrence (2013), and heavy-tailed priors Shah et al. (2014); Chen et al. (2020). State-space/Markov models posit a latent process with transition and observation mechanisms, enabling Bayesian filtering, smoothing, and forecasting Durbin & Koopman (2012); Rabiner (2002). SDE models represent continuous-time dynamics via discretized likelihoods or simulation and natural handling of irregular sampling Øksendal (2003). While effective under well-specified priors, these approaches remain constrained by the chosen prior family and can be sensitive to prior misspecification.

Recent studies combine traditional priors with neural networks to increase representational flexibility while retaining probabilistic structure Wilson et al. (2016); Sendera et al. (2021); Dutordoir

et al. (2018); Maroñas et al. (2021); Yu et al. (2021). Deep kernel learning (DKL) parameterizes GP kernels with neural feature maps, alleviating manual kernel engineering Wilson et al. (2016). Flow-based GP augmentations transform GP posterior distributions to capture non-Gaussian, input-dependent uncertainty (e.g., NGGP Sendera et al. (2021) and CNF-DGP Yu et al. (2021)). Neural SDEs parameterize the drift and diffusion coefficients with neural networks, extending SDE-based model to richer continuous-time dynamics and accommodating irregular sampling Tzen & Raginsky (2019). Despite these advances, learning remains anchored to predefined prior scaffolds (e.g., kernel families, GP priors, SDE structure), and many methods—such as flow-based GP augmentations and neural SDEs—often rely on multi-trajectory supervision and entail substantial training costs. Different from prior-driven methods, DBPT, as a "shapeshifter", uses a weak-prior and attains greater representational flexibility.

**Data-Driven Approaches.** Data-driven methods typically dispense with explicit structural priors and learn conditional stochastic processes via neural networks and amortized inference across multiple trajectories. Conditional neural processes (CNPs) use a permutation-invariant set encoder to summarize the context set and a decoder to produce predictive distributions for query inputs, enabling fast adaptation Garnelo et al. (2018b;a). Convolutional CNPs Gordon et al. (2019) and Relational CNPs Huang et al. (2023) integrate a broad class of group equivariances into NP architectures. Other variants introduce mechanisms such as autoregressive prediction Bruinsma et al. (2023), contrastive training Ye & Yao (2022), and hierarchical evidential distributions Pandey & Yu (2023), among others. Although these methods can be trained on a single trajectory via episodic segmentation, their uncertainty estimates tend to be less reliable and their advantages often diminish. Different from NPs, which amortize across multi-trajectory datasets and may suffer amortization gaps and miscalibration in the single-trajectory regime, DBPT hard-codes projective consistency and learns from a single trajectory.

**Conditional Generative Models** Beyond stochastic-process methods, our paradigm is conceptually related to modern generative models—such as normalizing flows Papamakarios et al. (2021) and diffusion models Croitoru et al. (2023)—that learn conditional laws $p(x_s \mid s)$ by transporting base noise at the instance level (i.e., separately for each $s \in \mathcal{T}$). These models usually need multiple samples per index $s$ to learn $p(x_s \mid s)$ effectively. Moreover, because $s$ is treated as a fixed conditioning variable, they do not capture dependencies across $s_1, \ldots, s_n$ and thus do not induce a process-level joint distribution. In contrast, DBPT defines and learns a process-level pushforward on path space and ensure projective consistency across finite marginals. The advantages of our approach are summarized in Appendix E.

## 4 EXPERIMENT

In this section, we evaluate the performance of the DBPT across several widely used benchmark, including synthetic trajectory, time series, image completion, and black-box optimization to assess its effectiveness. We compare the results of DBPT against several baseline and state-of-the-art (SOTA) methods commonly used in the field, such as standard Gaussian processes (GPs) Wilson et al. (2011), warped Gaussian processes (WGP) Lázaro-Gredilla (2012), Markov model Rabiner (2002), deep kernel learning (DKL) Wilson et al. (2016), SDE matching (a kind of neural SDE) Bartosh et al. (2025), and conditional neural processes (CNP) Garnelo et al. (2018a). Specifically, since our focus is single-trajectory scenarios, all experiments in this section are conducted within a single-trajectory data. For the multi-trajectory methods (SDE matching, and CNP), we train them using a single trajectory via episodic segmentation. All methods are evaluated with the same sampling budget and protocol to ensure a fair comparison. Detailed descriptions of the experimental configurations and results are provided in the Appendix F.

### 4.1 VISUALIZATION OF SYNTHETIC TASK

To showcase DBPT's flexibility and adaptability over alternative approaches, we construct two synthetic datasets, each reflecting a different stochastic modeling scenario: a Gaussian-process dataset with a smooth kernel and a Markov-process dataset exhibiting temporal dependence. See the Appendix G for a detailed description. Figure 2 present the visual experimental results.

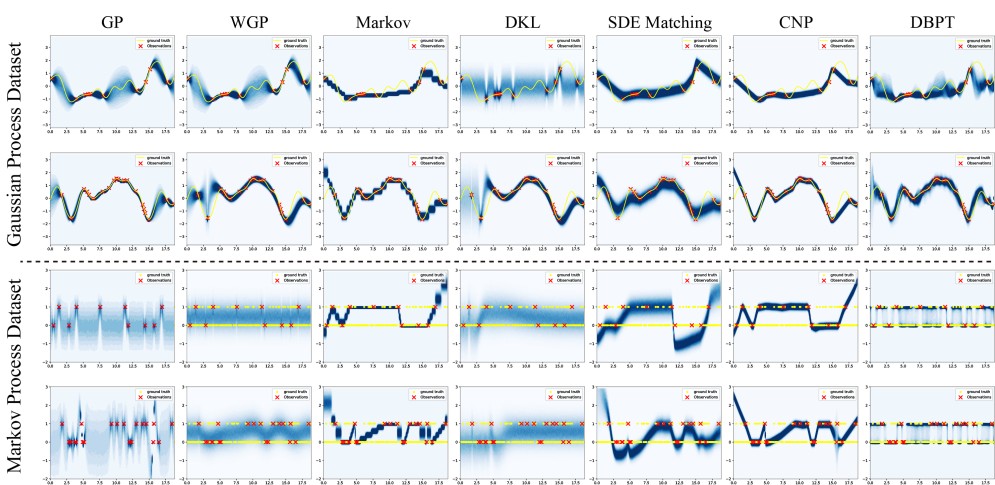

Figure 2: The results on synthetic data generated from the Gaussian process and Markov process. For each dataset, observations are set at positions [10, 20]. The red markers indicate the observations, the blue regions represent the estimating distribution of the methods, and the yellow lines indicate the ground truth.

On the Gaussian process dataset, GP demonstrate superior performance, as their priors are highly consistent with the target data. However, when the prior information is mismatched, such as when applying GP to the Markov process dataset, performance degrades. In comparison of GPs, the Markov-based method behaves in the opposite manner at different task.This is because prior-driven methods remain constrained by the chosen prior family and can be sensitive to prior misspecification. For WGP, when the number of observation points is limited, it struggles to construct an accurate distribution. We observe that NGGP struggles to converge on single-trajectory data. CNP, when trained on a single-trajectory data, suffers from poorly calibrated uncertainty,thereby limiting extrapolative performance and reliability. Compared to other methods, DBPT exhibits robust adaptability on both Gaussian and Markov data, owing to its deconvolution-based process transformation, which can automatically accommodate arbitrary problem structures and mitigate the dependence on prior. The synthetic experiment demonstrate that DBPT offers superior flexibility and adaptability, enabling the modeling of complex and diverse stochastic processes. The gains are consistent with our by-design consistency: generating the entire trajectory in one pass avoids post-hoc stitching of marginals and preserves cross-index dependencies.

## 4.2 Time Series Modeling

Time series data typically exhibit randomness and complex dependencies that evolve over time. In this section, we evaluate the performance of the proposed algorithm through experiments conducted on finance dataset. The dataset consists of the daily closing prices from January 1, 2024, to December 31, 2024, provided by Pudong Development Bank (PDB) and Guangzhou Baiyun International Airport (BIA) in China A-shares. Table 1 presents the experimental result.

On complex financial datasets, GPs underperform because their priors are not easily adaptable for learning flexible representations. Although WGP underperforms DBPT on the PDB task, it still achieves the best average performance, owing to its output-space warping, which makes it better suited to such highly complex problems. The Markov and CNP methods perform well in terms of MSE, largely because their low-variance predictions reduce the discrepancy between predicted and actual values. However, this significantly impairs the model's ability to accurately capture and represent the uncertainty associated with the target points, causing worse performance in terms of NLL. In essence, the model's tendency to provide overly confident predictions diminishes its ability to handle the inherent uncertainty of the data. SDE Matching performs poorly on this type of task. On the complex financial dataset, DBPT attains the second-best average performance, trailing only WGP. DBPT places a stronger emphasis on modeling the uncertainty of target points, this focus comes at the cost of lower MSE performance compared to CNP. The model introduces more

Table 1: Performance comparison on finance benchmarks. Lower value indicates better result. Best means are highlighted in bold. We report Mean ± Std and the average rank over four metrics.

| | BIA ↓ | | PDB ↓ | | Avg. Rank |
| | NLL | MSE | NLL | MSE | |
|---|---|---|---|---|---|
| GP | 798.49 ± 77.06 | 18.63 ± 13.18 | 686.58 ± 45.04 | 11.71 ± 5.28 | 5.00 |
| WGP | **602.42** ± 55.42 | **4.12** ± 1.17 | 504.32 ± 25.98 | 2.34 ± 0.36 | **1.75** |
| MARKOV | 724.21 ± 68.68 | 5.00 ± 3.52 | 695.33 ± 115.15 | **1.63** ± 0.81 | 3.00 |
| DKL | 1005.31 ± 19.02 | 28.47 ± 4.83 | 760.00 ± 30.04 | 8.01 ± 1.39 | 5.75 |
| SDE Matching | 2130.04 ± 256.41 | 85.85 ± 32.25 | 1681.99 ± 296.93 | 61.51 ± 19.90 | 7.00 |
| CNP | 686.82 ± 45.70 | 7.07 ± 2.41 | 509.43 ± 53.62 | 1.97 ± 0.28 | 3.00 |
| DBPT | 647.92 ± 135.30 | 5.98 ± 2.91 | **501.00** ± 36.07 | 3.40 ± 0.8 | 2.50 |

Table 2: Quantitative evaluation on image completion using PSNR and SSIM (higher is better). For each benchmark the best result is highlighted in bold. The last column reports the average rank.

| | MNIST ↑ | | CIFAR ↑ | | |
| | PSNR | SSIM | PSNR | SSIM | Avg. Rank |
|---|---|---|---|---|---|
| GP | 6.33 ± 0.1 | 0.01 ± 0.0 | 10.57 ± 0.03 | 0.05 ± 0.0 | 5.50 |
| WGP | 6.41 ± 0.16 | 0.02 ± 0.0 | 10.62 ± 0.08 | 0.06 ± 0.0 | 4.50 |
| MARKOV | 13.9 ± 1.19 | 0.62 ± 0.06 | 14.86 ± 0.26 | 0.38 ± 0.07 | 2.75 |
| DKL | 6.76 ± 0.35 | 0.11 ± 0.01 | 15.88 ± 1.19 | 0.45 ± 0.05 | 4.00 |
| CNP | 16.58 ± 0.57 | 0.62 ± 0.04 | 18.56 ± 0.35 | 0.61 ± 0.04 | 2.00 |
| DBPT | **21.65** ± 1.32 | **0.94** ± 0.02 | **24.04** ± 2.50 | **0.9** ± 0.06 | **1.00** |

variability in its predictions to better capture the uncertainty, which increases the prediction error for individual points and results in a higher MSE. However, this trade-off allows DBPT to more accurately estimate the underlying data distribution, leading to significantly better performance in terms of NLL.

### 4.3 IMAGE COMPLETION

In the image, each pixel can be seen as a random variable, where its value is inherently dependent on its two-dimensional spatial indices. The spatial dependencies and the underlying image structure dictate that this stochastic process evolves across both dimensions. We evaluate the performance of the algorithms on the CIFAR and MNIST datasets for image completion. During training, we randomly mask a portion of the pixels, treating it as a single-trajectory image completion problem. Given the significantly high computational cost of SDE matching, we did not include it in the comparisons in this experiments. More details and results are provided in the Appendix H.

Figure 3 presents the visual results and table 2 presents the quantitative result using PSNR and SSIM. Due to the strong prior, both GP, WGP, and Markov model struggle to model the target problem effectively, resulting in highly blurred images on both MNIST and CIFAR. In contrast, DKL leverages neural networks to learn flexible representations and therefore achieves substantial gains over standard GP baselines (both qualitatively and quantitatively). However, its completions often contain artifacts—e.g., hallucinated or non-existent colors relative to the source image. CNP produces sharper images on CIFAR, likely because in the single-trajectory setting it overfits to the target data and thus underestimates uncertainty; the resulting sampling process is less stochastic, yielding crisper outputs. Nevertheless, similar to DKL, CNP frequently exhibits color shifts and localized reconstruction errors (e.g., miscompleted horse legs), attributable to insufficient uncertainty. By contrast, DBPT delivers the best overall results on image completion—qualitatively and quantitatively—demonstrating strong representational flexibility together with reliable uncertainty quantification in small-sample, single-trajectory regimes.

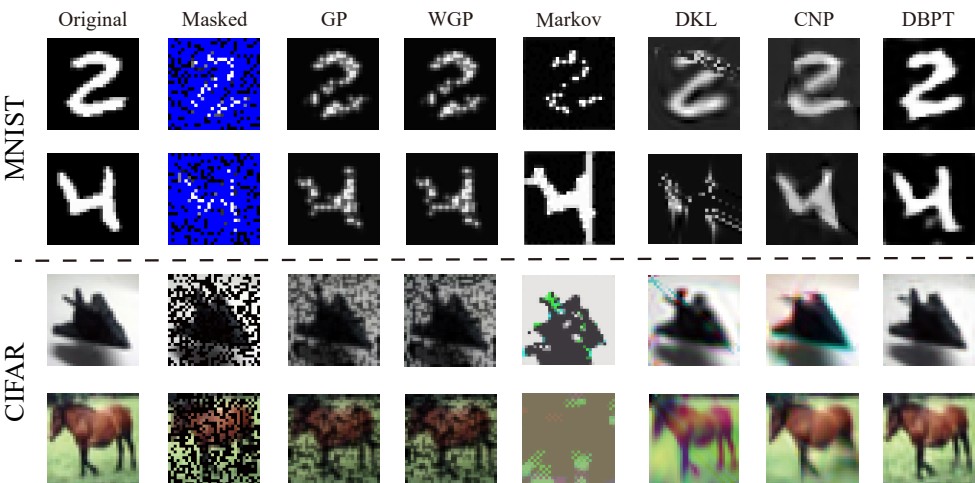

Figure 3: Image completion on the MNIST and CIFAR datasets. The "Masked" columns show the masked images, with unmasked pixels treated as known observations. For MNIST, the masked points are represented in blue, while for CIFAR, they are represented in black.

## 4.4 BLACK-BOX OPTIMIZATION

In this experiment, we systematically compare the performance from a black-box optimization (BBO) perspective. BBO typically involves evaluating an objective function that is difficult to express explicitly or compute analytically, only obtaining a single-trajectory data. Bayesian optimization (BO) addresses BBO by employing GPs to probabilistically model the landscape of the objective function, leveraging uncertainty estimates to guide the search process. We integrate compared stochastic process modeling methods as surrogate model into the Bayesian optimization framework and choose the expected improvement acquisition function. We validate the results on two widely used multimodal black-box optimization problems, Schwefel and Rastrigin. Detailed information can be found in the Appendix I.

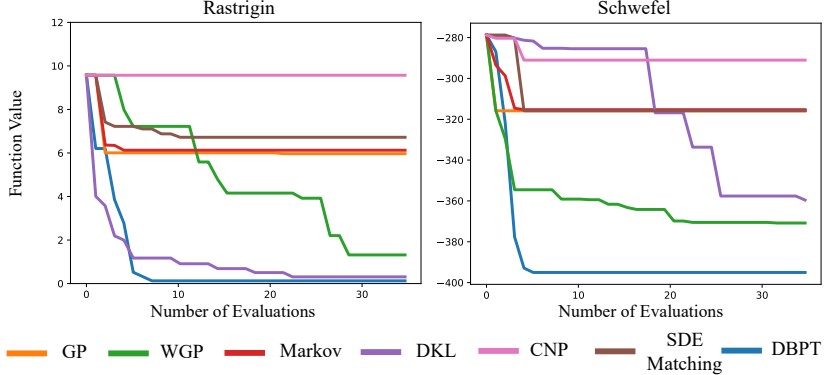

Figure 4: Optimization results (lower is better) on the Schwefel and Rastrigin problems, presented as averaged convergence curves, where DBPT outperforms the other methods.

Figure 4 presents the optimization results of comparative algorithms. The Schwefel and Rastrigin problems exhibit global structures with varying levels of complexity, and thus the prior of GP and Markov model limits their ability to adapt to such complex landscapes, resulting in poor performance. WGP demonstrates sustained optimization potential on both tasks, which may be due to its nonlinear transformation enhancing the adaptability of GP. CNP, due to overfitting the target problem in the single-trajectory setting, thus fails to provide sufficient uncertainty information to

guide the search process and leading to suboptimal results. DKL shows strong performance on the Rastrigin problem, likely benefiting from its increased representational flexibility, whereas Neural SDE underperforms. Since DBPT uses weaker prior, it exhibits greater flexibility and adaptability on more complex tasks. DBPT can model the landscape distribution of the entire problem more accurately and provide high-confidence uncertainty estimates, enabling it to find better solutions with fewer evaluations.

### 4.5 ABLATION AND PARAMETER ANALYSIS

We first examine how *output-space grid resolution* affects trajectory smoothness. We uniformly discretize the domain into a base grid with $N = 200$ points and vary the resolution $N \in \{200, 400, 600, 800\}$. As shown in Fig. 5, increasing $N$ beyond 400 yields diminishing returns in reconstruction fidelity while amplifying high-frequency noise and degrading the calibration of predictive uncertainty. Higher resolutions also cause the model to overemphasize local fluctuations, producing markedly jagged trajectories. These results suggest that a modest resolution—about $1\times$–$2\times$ the base grid (i.e., $N = 200$–$400$)—strikes a favorable balance among fidelity, smoothness, and efficiency. We also perform an ablation on the architecture. See more details in the Appendix J.

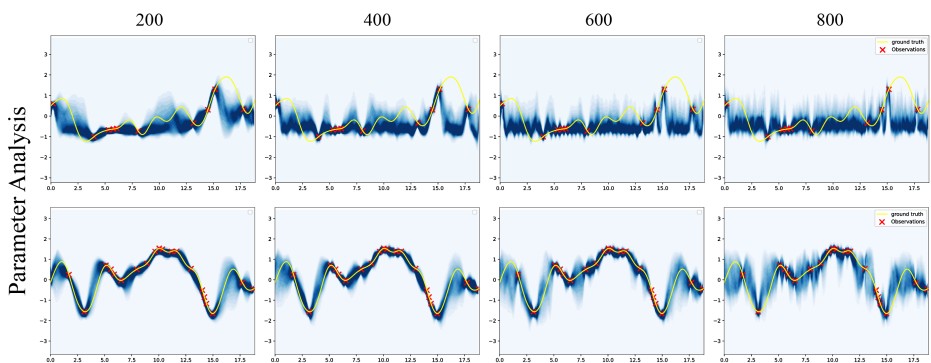

Figure 5: Parameter analysis of the output-space grid resolution on synthetic data.

## 5 CONCLUSION

We introduce a stochastic-process modeling paradigm that learns a noise-to-process transform and instantiate it with a deconvolution-based architecture (DBPT) to capture inter-temporal dependence for weak-prior, single-trajectory regimes. On the theory side, we formalize a learnable N2P representation in which a shared base-noise process is mapped by a single measurable generator to a full trajectory; this structure makes finite-index marginals projections of the same joint sample, rendering projective consistency intrinsic by design and compatible with Kolmogorov extension to denser index sets. Empirically, DBPT retains the data efficiency of prior-driven methods and the flexibility of representations—without multi-trajectory supervision—while achieving competitive performance across diverse single-trajectory benchmarks with sparse observations.

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

APPENDIX

# A  CONSTRUCTIONS

## A.1  COUNTABLE INDEX SETS (ROSENBLATT-TYPE REALIZATION)

Let $\mathcal{T}$ be countable with enumeration $t_1, t_2, \ldots$ and $Y_k := Y(t_k)$. Let the base noise be $Z_i \sim \mathrm{Unif}(0, 1)$ i.i.d. (by base-noise invariance below, this entails no loss of generality).

**Proposition 5** (Rosenblatt-type factorization). *There exist Borel measurable maps $\psi_1 : [0, 1] \to \mathsf{S}$ and $\psi_k : \mathsf{S}^{k-1} \times [0, 1] \to \mathsf{S}$ $(k \geq 2)$ such that the recursion $\tilde{Y}_1 = \psi_1(Z_1)$, $\tilde{Y}_k = \psi_k(\tilde{Y}_1, \ldots, \tilde{Y}_{k-1}, Z_k)$ satisfies $(\tilde{Y}_1, \ldots, \tilde{Y}_n) \stackrel{d}{=} (Y_1, \ldots, Y_n)$ for all $n$.*

*Idea.* Use existence of regular conditional distributions on standard Borel spaces and the (generalized) probability integral transform to push $\mathrm{Unif}(0, 1)$ to each conditional law, with joint measurability ensured via a Borel isomorphism $b : \mathsf{S} \to (0, 1)$ and generalized inverses of conditional cdfs. Independence of $(Z_i)$ and induction complete the proof.

Let $F : [0, 1]^{\mathbb{N}} \to \mathsf{S}^{\mathbb{N}}$ map $(Z_i)$ to $(\tilde{Y}_i)$. Then the first $n$ coordinates of $F(Z)$ match $(Y_1, \ldots, Y_n)$ for each $n$, hence the fidim laws agree.

## A.2  PATH-SPACE REALIZATION (POLISH FUNCTION SPACES)

If $Y(\cdot)$ admits a version in a Polish path space $\mathcal{E}$ (e.g., $C(K; \mathbb{R}^{d_y})$ or $D([0, 1]; \mathbb{R}^{d_y})$), then:

**Proposition 6** (Path-space factorization). *There exists a Borel measurable $T : [0, 1] \to \mathcal{E}$ such that $T(U) \sim \mathsf{P}_{Y(\cdot)}$ for $U \sim \mathrm{Unif}(0, 1)$. Writing $G(x, z) = T(z)(x)$ yields $Y(x) \stackrel{d}{=} G(x, U)$.*

## A.3  BASE-NOISE INVARIANCE

**Proposition 7** (Base-noise invariance). *If $(Z_i)$ and $(Z_i')$ are i.i.d. with non-atomic laws on standard Borel spaces, then there exists a Borel isomorphism $\Psi$ such that $Z_i' = \Psi(Z_i)$ a.s. for all $i$. Consequently any representation $H((Z_i))$ may be transferred to $H((\Psi^{-1}(Z_i')))$.*

**Remark on terminology.** In modern texts (e.g., Dudley, Kallenberg), exchangeability and consistency correspond to *permutation consistency* and *projective consistency*; the latter is precisely Proposition 11.

# B  FOUNDATIONAL PROPERTIES OF THE LEARNABLE N2P REPRESENTATION

**Setup.** Let the index set be $\mathcal{T}$ and the state space $\mathsf{S} = \mathbb{R}^{d_y}$ with its Borel $\sigma$-algebra. Let $(\mathcal{Z}, \mathcal{B}(\mathcal{Z}))$ be a standard Borel space and let $Z = (Z_i)_{i \geq 1} \sim \nu$ denote an i.i.d. *shared base-noise process* with product law $\nu$. A (pathwise) generator is a measurable map $G_\theta : \mathcal{Z}^{\mathbb{N}} \to \mathsf{S}^{\mathcal{T}}$, and we write

$$X = G_\theta(Z) \in \mathsf{S}^{\mathcal{T}}, \qquad \mu_\theta = \nu \circ G_\theta^{-1} \in \mathcal{P}(\mathsf{S}^{\mathcal{T}}).$$

### B.1 MEASURABILITY AND PROCESS VALIDITY

**Lemma 8** (Measurability of evaluation). *Let* $\mathrm{ev}_I : \mathsf{S}^{\mathcal{T}} \to \mathsf{S}^I$ *be the coordinate projection for finite* $I \subset \mathcal{T}$. *Then* $\mathrm{ev}_I$ *is Borel measurable.*

*Proof.* By definition of the cylinder $\sigma$-algebra on $\mathsf{S}^{\mathcal{T}}$, each coordinate map is measurable, hence the finite product $\mathrm{ev}_I$ is measurable.

**Lemma 9** (Functoriality of pushforwards). *For measurable* $\psi, \phi$ *and measure* $\eta$, $(\phi \circ \psi)_{\#}\eta = \phi_{\#}(\psi_{\#}\eta)$.

**Proposition 10** (Well-defined process on $\mathcal{T}$). *Let* $\mu_{\theta} = \nu \circ G_{\theta}^{-1}$. *Then* $X \sim \mu_{\theta}$ *is a well-defined stochastic process on* $\mathcal{T}$, *and for any finite* $I \subset \mathcal{T}$ *the finite-dimensional law is*

$$\mu_{\theta, I} := (\mathrm{ev}_I)_{\#}\,\mu_{\theta} \in \mathcal{P}(\mathsf{S}^I).$$

**Proposition 11** (Intrinsic projective consistency). *For* $J \subset I \subset \mathcal{T}$, $\pi_J \# \mu_{\theta, I} = \mu_{\theta, J}$, *where* $\pi_J : \mathsf{S}^I \to \mathsf{S}^J$ *is the coordinate projection.*

*Proof.* By Lemma 8, $\mathrm{ev}_J = \pi_J \circ \mathrm{ev}_I$. Applying Lemma 9, $\mu_{\theta, J} = (\mathrm{ev}_J)_{\#}\mu_{\theta} = (\pi_J \circ \mathrm{ev}_I)_{\#}\mu_{\theta} = \pi_J \#((\mathrm{ev}_I)_{\#}\mu_{\theta}) = \pi_J \# \mu_{\theta, I}$.

**Corollary 12** (Permutation consistency). *For any permutation* $\sigma \in S_n$ *and ordered* $X = (x_1, \ldots, x_n)$, *letting* $P_{\sigma} : \mathsf{S}^n \to \mathsf{S}^n$ *be the coordinate permutation,* $\mu_{X_{\sigma}} = (P_{\sigma})_{\#}\,\mu_X$.

### B.2 COMPATIBILITY WITH KOLMOGOROV EXTENSION

Let $\{\mathcal{T}_n\}_{n \geq 1}$ be a nondecreasing sequence of finite or countable grids with $\bigcup_n \mathcal{T}_n$ dense in a continuum index set. Define $\mu_{\theta}^{(n)}$ as the marginal of $\mu_{\theta}$ on $\mathsf{S}^{\mathcal{T}_n}$.

**Corollary 13** (Compatibility with Kolmogorov). *The family* $\{\mu_{\theta}^{(n)}\}_{n \geq 1}$ *is pairwise projectively consistent by Proposition 11, hence by Kolmogorov's extension theorem (for standard Borel state spaces) there exists a probability measure on* $\mathsf{S}^{\bigcup_n \mathcal{T}_n}$ *with these marginals. This is a compatibility statement stemming from the N2P structure; no additional modeling assumption is required.*

## C GENERALIZATION AND CALIBRATION UNDER RANDOM MASKS

We work with the learnable noise-to-process (N2P) representation $X = G_{\theta}(Z)$ on a finite grid $\mathcal{T} = \{t_1, \ldots, t_{|\mathcal{T}|}\}$, where a *single* measurable generator $G_{\theta} : \mathcal{Z}^{\mathbb{N}} \to \mathsf{S}^{\mathcal{T}}$ maps a shared i.i.d. base-noise process $Z$ to a full trajectory $X = (X_t)_{t \in \mathcal{T}}$; this yields a valid process law and intrinsic projective consistency (see main-text Def. 1 and Props. 2–3).

### C.1 SETUP AND SAMPLING MODEL

Let $\mathsf{S} = \mathbb{R}^{d_y}$ and let $Z$ be an i.i.d. base-noise sequence with product law $\nu$ on a standard Borel space $\mathcal{Z}$. For any parameter $\theta$, denote a path sample on the grid by

$$\widehat{X}_{\theta}(\mathcal{T}; Z) = G_{\theta}(Z) \in \mathbb{R}^{|\mathcal{T}| \times d_y}, \qquad \widehat{X}_{\theta}(t; Z) \in \mathbb{R}^{d_y}.$$

**Random-mask (mini-batch) training protocol on a *fixed* observation set.** In our datasets, only a fixed subset of indices $\tau_o \subset \mathcal{T}$ is observed. At each iteration we draw independently: (i) a base-noise sequence $Z \sim \nu$; (ii) an *optional mini-batch mask* $M = (M_t)_{t \in \mathcal{T}}$ supported on $\tau_o$, with $M_t \overset{\text{i.i.d.}}{\sim} \mathrm{Bernoulli}(q)$ for $t \in \tau_o$ and $M_t \equiv 0$ for $t \notin \tau_o$, independent of $Z$. Only coordinates with $M_t = 1$ are penalized in that iteration. Let $X^{\star} : \mathcal{T} \to \mathbb{R}^{d_y}$ be the (deterministic) target trajectory on $\mathcal{T}$.

We define the *population* masked MSE on $\tau_o$ and its *unbiased* mini-batch estimator as

$$\mathcal{L}_{\tau_o}(\theta) := \mathbb{E}_Z\left[\frac{1}{|\tau_o|}\sum_{t\in\tau_o}\left\|\widehat{X}_\theta(t;Z) - X^\star(t)\right\|^2\right], \tag{1}$$

$$\widehat{\mathcal{L}}_{\tau_o,n}(\theta) := \frac{1}{n}\sum_{k=1}^n\left\{\frac{1}{q|\tau_o|}\sum_{t\in\tau_o}M_t^{(k)}\left\|\widehat{X}_\theta(t;Z^{(k)}) - X^\star(t)\right\|^2\right\}, \tag{2}$$

where $\{(Z^{(k)}, M^{(k)})\}_{k=1}^n$ are i.i.d. and $M^{(k)}$ is supported on $\tau_o$. Clearly $\mathbb{E}_M[\frac{1}{q|\tau_o|}\sum_{t\in\tau_o}M_t\cdot] = \frac{1}{|\tau_o|}\sum_{t\in\tau_o}\cdot$, hence $\mathbb{E}[\widehat{\mathcal{L}}_{\tau_o,n}(\theta)] = \mathcal{L}_{\tau_o}(\theta)$.

**Remark 14** (Effective sampling view). *Each iteration contributes an expected $q|\tau_o|$ penalized coordinates. Thus $n$ iterations correspond to an effective sample size $N = n q|\tau_o|$ for generalization analysis.*

## C.2 Uniform convergence on a finite grid

**Assumption 15** (Uniform boundedness and Lipschitz in noise). *There exist constants $B, L_Z < \infty$ such that for all $\theta$, all $Z$, and all $t \in \mathcal{T}$, $\|\widehat{X}_\theta(t;Z)\| \leq B$, and for any $Z, Z'$, $\big\|\widehat{X}_\theta(\mathcal{T};Z) - \widehat{X}_\theta(\mathcal{T};Z')\big\|_F \leq L_Z\|Z - Z'\|$, where $\|\cdot\|$ is a product metric on $\mathcal{Z}^{\mathbb{N}}$ (e.g., a summably weighted $\ell_2$ metric) under which $\mathbb{E}\|Z\|^2 < \infty$.*

**Definition 16** (Function class and Rademacher complexity). *Let $\mathcal{F} := \big\{(t,Z) \mapsto \langle e, \widehat{X}_\theta(t;Z)\rangle : \|e\|_2 = 1, \theta \in \Theta\big\}$. Let $\{(t_i, Z_i)\}_{i=1}^N$ be i.i.d. with $t_i$ uniform on $\tau_o$ and $Z_i \sim \nu$ independent. The empirical Rademacher complexity is $\widehat{\mathfrak{R}}_N(\mathcal{F}) := \mathbb{E}_\sigma\big[\sup_{f\in\mathcal{F}}\frac{1}{N}\sum_{i=1}^N\sigma_i f(t_i, Z_i)\big]$, and $\mathfrak{R}_N(\mathcal{F}) = \mathbb{E}\big[\widehat{\mathfrak{R}}_N(\mathcal{F})\big]$.*

**Lemma 17** (Mask reduction (unbiased mini-batch)). *Fix $\theta$ and $n$. With probability at least $1 - \delta$ over $\{M^{(k)}\}_{k=1}^n$,*

$$\left|\widehat{\mathcal{L}}_{\tau_o,n}(\theta) - \frac{1}{nq|\tau_o|}\sum_{k=1}^n\sum_{t\in\tau_o}\left\|\widehat{X}_\theta(t;Z^{(k)}) - X^\star(t)\right\|^2\right| \leq B^2\sqrt{\frac{2\log(2/\delta)}{nq|\tau_o|}},$$

*and $\mathbb{E}_M\big[\widehat{\mathcal{L}}_{\tau_o,n}(\theta)\big] = \frac{1}{n}\sum_{k=1}^n\frac{1}{|\tau_o|}\sum_{t\in\tau_o}\|\widehat{X}_\theta(t;Z^{(k)}) - X^\star(t)\|^2$.*

*Proof.* Conditioned on $\{Z^{(k)}\}$, the inner sum is a Bernoulli subsampling average of bounded terms on $\tau_o$; apply Hoeffding/McDiarmid with $\mathbb{E}[\sum_{t\in\tau_o}M_t] = q|\tau_o|$, then average over $k$.

**Theorem 18** (Uniform convergence under mini-batch masks). *Under Assumption 15, for any $\delta \in (0,1)$, with probability at least $1 - \delta$ over $\{(Z^{(k)}, M^{(k)})\}_{k=1}^n$,*

$$\sup_\theta\left|\mathcal{L}_{\tau_o}(\theta) - \widehat{\mathcal{L}}_{\tau_o,n}(\theta)\right| \leq C_1 B^2\sqrt{\frac{\log(4/\delta)}{nq|\tau_o|}} + 2\mathfrak{R}_{nq|\tau_o|}(\mathcal{F}) + c\frac{B^2\log(4/\delta)}{nq|\tau_o|},$$

*for absolute constants $C_1, c$. Moreover, if $G_\theta$ is uniformly $L_Z$-Lipschitz in $Z$ and $\mathbb{E}\|Z\|^2 < \infty$, then by vector contraction*

$$\mathfrak{R}_N(\mathcal{F}) \leq C_2\frac{L_Z\big(\mathbb{E}\|Z\|^2\big)^{1/2}}{\sqrt{N}}, \qquad N = nq|\tau_o|,$$

*hence $\sup_\theta|\mathcal{L}_{\tau_o}(\theta) - \widehat{\mathcal{L}}_{\tau_o,n}(\theta)| = \tilde{O}\big(\sqrt{\frac{1}{nq|\tau_o|}} + \frac{1}{\sqrt{n}}\big)$.*

# D IDENTIFIABILITY: PATH LEVEL VS. PROCESS LAW (UP TO NOISE ISOMORPHISM)

We distinguish *path identifiability* (recovering the realized trajectory under regularity) from *process-law identifiability* (recovering the joint law). The N2P structure ensures a valid law on $S^{\mathcal{T}}$ and intrinsic projective consistency (main-text Props. 2–3), and is compatible with Kolmogorov extension on nested grids.

### D.1 PATH IDENTIFIABILITY UNDER HÖLDER REGULARITY

**Proposition 19** (Path identifiability under dense random masks). *Let $K \subset \mathbb{R}^d$ be compact and $X^\star \in C^\alpha(K)$ with $\alpha \in (0,1]$. Suppose $\{\tau_o^{(k)}\}_{k \geq 1}$ are i.i.d. random sample sets with densities bounded below on $K$ and $|\tau_o^{(k)}| \to \infty$. Then with probability 1, the union $\bigcup_{k \geq 1} \tau_o^{(k)}$ is dense in $K$, and $X^\star$ is the unique $C^\alpha$-continuous function consistent with all observations. If the model class $\{G_\theta\}$ is a universal approximator on $C^\alpha(K)$ and empirical risks vanish along a minimizing sequence, then any $L^2(\rho)$-limit point of $\widehat{X}_\theta$ equals $X^\star$.*

*Proof.* Since $\rho_{\min} > 0$, for any open ball $B \subset K$ we have $\mathbb{P}(B \cap \tau_o^{(k)} \neq \emptyset) \geq c > 0$, so by the Borel–Cantelli lemma $B$ is hit infinitely often almost surely; hence $\bigcup_k \tau_o^{(k)}$ is dense. Hölder regularity implies uniform continuity, so a continuous function is uniquely determined by its values on a dense set. Universal approximation and vanishing empirical risk yield $L^2(\rho)$ convergence to the unique continuous extension, which coincides with $X^\star$. $\square$

**Remark 20** (Optimization/statistics trade-off). *The conclusion assumes $\eta_m \to 0$ (negligible optimization error) and some capacity control (e.g., regularization or early stopping). Without such control, universal approximators can interpolate finite samples without guaranteeing convergence to $X^\star$.*

### D.2 PROCESS-LAW IDENTIFIABILITY UP TO NOISE ISOMORPHISM

**Definition 21** (Noise-isomorphism equivalence). *Two generators $G_\theta, G_{\theta'} : \mathcal{Z}^{\mathbb{N}} \to \mathsf{S}^{\mathcal{T}}$ are noise-isomorphism equivalent if there exists a measure-preserving Borel isomorphism $\psi : \mathcal{Z} \to \mathcal{Z}$ such that, for $Z = (Z_i)_{i \geq 1}$ i.i.d. with law $\nu$ and $\Psi(Z) := (\psi(Z_i))_{i \geq 1}$,*

$$G_{\theta'}(Z) \stackrel{d}{=} G_\theta(\Psi(Z)).$$

**Theorem 22** (Process-law identifiability up to noise isomorphism). *Let $Y$ be a target process law on $\mathsf{S}^{\mathcal{T}}$ whose finite-dimensional marginals are realized by some generator $G_{\theta^\star}$, i.e., $\mu_{\theta^\star, I}$ for all finite $I \subset \mathcal{T}$. If another generator $G_{\widehat{\theta}}$ satisfies $\mu_{\widehat{\theta}, I} = \mu_{\theta^\star, I}$ for all finite $I$, then $G_{\widehat{\theta}}(Z) \stackrel{d}{=} G_{\theta^\star}(Z)$, so they induce the same process law on $\mathsf{S}^{\mathcal{T}}$. Moreover, there exists a measure-preserving isomorphism $\psi$ on $\mathcal{Z}$ such that $G_{\widehat{\theta}}(Z) \stackrel{d}{=} G_{\theta^\star}(\Psi(Z))$.*

*Proof.* Equality of all finite-dimensional marginals implies equality of the cylinder set measures; by Kolmogorov's extension theorem (for standard Borel state spaces), the induced probability measures on $(\mathsf{S}^{\mathcal{T}}, \mathcal{C})$ coincide, so $G_{\widehat{\theta}}(Z) \stackrel{d}{=} G_{\theta^\star}(Z)$. For the isomorphism statement, note that any two non-atomic standard Borel probability spaces are measure-isomorphic; applying such an isomorphism coordinatewise to the i.i.d. sequence $Z$ preserves independence and the product law, yielding $\Psi(Z)$. Base-noise invariance then transfers the representation, giving $G_{\widehat{\theta}}(Z) \stackrel{d}{=} G_{\theta^\star}(\Psi(Z))$. $\square$

**Remark 23** (Maximal identifiable granularity). *With single-trajectory supervision, the maximal identifiable object at the process level is the equivalence class of generators modulo noise isomorphisms. Path-level identifiability (Prop. 19) characterizes the realized trajectory under regularity and dense observation, whereas process-law identifiability (Thm. 22) holds only up to noise isomorphism.*

### D.3 COMPATIBILITY WITH THE LEARNABLE N2P TRAINING PROTOCOL

Our $\tau_o$-masked objectives operate on a discrete grid and are fully consistent with N2P: the same $G_\theta$ and the same shared $Z$ produce all coordinates jointly; finite-dimensional marginals are intrinsic projections; and nested grids admit Kolmogorov-compatible extensions. These properties are independent of the training protocol.

# E  INNOVATION AND ADVANTAGES OF N2P PARADIGM

**Positioning.** The core contribution of this work is a computational methodology that *operationalizes* projective consistency for single-trajectory learning. Unlike GP/SDE approaches that rely on strong parametric priors and diffusion/flow models that typically condition pointwise, our *Noise-to-Process* design uses a single measurable generator that maps a shared noise process to full trajectories, ensuring provably coherent finite-dimensional marginals under sparse supervision. We instantiate this principle with DBPT, leveraging multi-scale deconvolution to propagate observation constraints and capture long-range dependencies under heavy missingness. The resulting model transfers across modalities (time series, images) and downstream tasks (e.g., black-box optimization) without task-specific likelihood heads, providing calibrated uncertainty and competitive accuracy with lightweight training in our settings. In short, we turn a consistency idea into a practical, unified recipe for learning process-level predictive distributions from limited data.

**Why N2P?**

- *Weak prior, strong flexibility.* No fixed kernel family, transition form, or SDE parameterization is imposed; dependence is learned via a pathwise map over trajectory space.

- *Single-trajectory supervision.* Training uses only masked reconstruction on observed indices, avoiding multi-trajectory or meta-datasets.

- *Process-level validity.* Modeling a pushforward measure on path space induces finite-dimensional laws that satisfy *projective (Kolmogorov) consistency* by construction.

- *Architecture-agnostic instantiation.* Our deconvolutional instance (DBPT) enforces multi-scale coherence, capturing both local and long-range dependencies while enabling flexible, distribution-level uncertainty.

- *Lightweight computation.* In our evaluations, training is lightweight and efficient, with competitive accuracy and calibrated uncertainty across modalities (time series, images) and downstream tasks (e.g., black-box optimization).

# F  EXPERIMENT AND ALGORITHM CONFIGURATIONS

**NLL.** At test time, we draw $S = 10{,}000$ posterior *trajectory* samples per case. For each index $i$, we form a piecewise-constant predictive density $\hat{p}_i(\cdot)$ from a histogram with $B$ bins and width $h$, using the *same bin boundaries for all methods and test cases* (precomputed from pooled validation samples). The NLL is the average negative log-density at the ground truth (with $\varepsilon = 10^{-12}$ to avoid $\log 0$):

$$\text{NLL} = -\frac{1}{N_{\text{test}}\,|\mathcal{I}|} \sum_{n=1}^{N_{\text{test}}} \sum_{i \in \mathcal{I}} \log\big(\hat{p}_i^{(n)}(y_i^{*(n)}) + \varepsilon\big).$$

The mean squared error (MSE) is computed at all unobserved evaluation points by comparing the predicted values against the ground truth and averaging across trajectories. For the synthetic and time-series benchmarks, each trial randomly assigns 10% of the data for training and the remaining 90% for evaluation; all experiments are repeated 10 times with independent splits. For all quantitative results, the synthetic, time-series, and black-box optimization tasks are repeated 10 times, whereas the image experiments are repeated 5 times due to their higher computational cost; we report the mean and standard deviation.

For all baselines, we follow the hyperparameter and architecture settings reported in the original papers. For DBPT, in Synthetic, time-series, and black-box optimization tasks, the noise encoder is a one-layer MLP. The deconvolution-based process decoder comprises three deconvolution blocks (each block performs one upsampling followed by one 1D convolution), and a final 1D convolution output layer. In image completion, the noise encoder consists of one 2D convolution layer followed by two MLP layers. The deconvolution-based process decoder uses two deconvolution blocks, and a final 2D convolution output layer. We use the Adam optimizer with a learning rate of $1 \times 10^{-3}$. All code (and training/evaluation scripts) will be made publicly available upon acceptance.

# G    SYNTHETIC TASK

The Gaussian Process and Markov Process are designed as follows:

- **Gaussian Process Dateset:**    $GP \sim (m(i), K(i, i')),$    $m(i) = 0,$
  $k(\mathbf{i}, \mathbf{i}') = \sigma^2 \exp\left(-\frac{\|\mathbf{i}-\mathbf{i}'\|^2}{2\ell^2}\right), \sigma = 1.0, \ell = 1.0, i \in [0, 6\pi]$

- **Markov Process Dateset:**    $\text{MP} \sim P(X),$
  $P(X) = \prod_{i=1}^{N} P(X_i \mid X_{i-1}) = \frac{1}{Z} \exp(-\sum_i \theta_i X_i - \sum_{i,j} \theta_{ij} X_i X_j), X(i) \in \{0, 1\}$

# H    DETAILS OF IMAGE COMPLETION

In the image completion experiments, we use 100 observed pixels for MNIST but increase this to 500 for CIFAR, since the RGB images are more complex and require more observations. Additional image results are provided in Fig. 7 (CIFAR) and Fig. 6 (MNIST). For both datasets, images are randomly sampled from each class (i.e., no cherry picking). We further observe that **DBPT** benefits from more observations: completion quality consistently improves as the fraction of observed pixels increases.

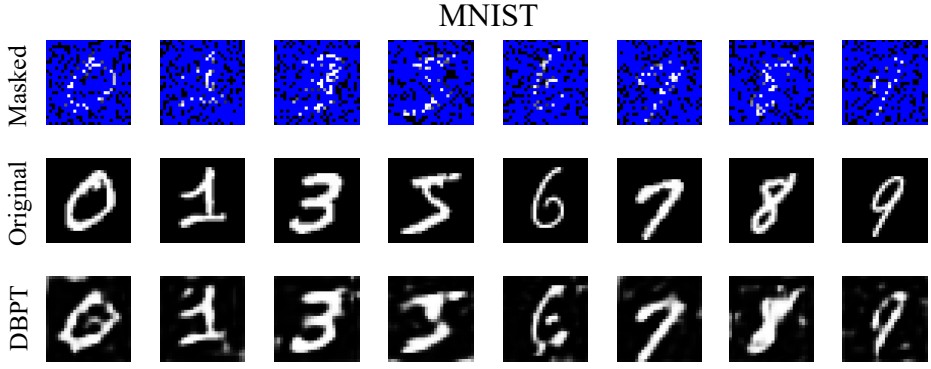

Figure 6: Image completion on MNIST dataset. The "Masked" columns show the masked images, with unmasked pixels treated as known observations The masked pixels are represented in blue.

# I    DETAILS OF BLACK-BOX OPTIMIZATION

Black-box Optimization (BBO) Lu et al. (2023) is a task that aims to find the global optimal solution for a given black-box problem. Unlike traditional optimization methods, which rely on an explicit mathematical model or gradient information of the objective problem, BBO treats the objective problem as a "black box". This means that the internal workings of the problem are not directly accessible or observable. Instead, we can only infer its behavior by providing inputs and receiving corresponding outputs. The black-box nature of the problem makes it difficult, or even impossible, to gather data for multi-trajectory datasets or to directly model the problem's internal mechanics. Mathematically, BBO problems are typically formulated as follows:

$$\arg\min f(\boldsymbol{x}), \quad \boldsymbol{x} \in \Omega \subset \mathbb{R}^n,$$

where $\boldsymbol{x}$ represents the decision variables, i.e., solutions, and $\Omega$ is the search domain in $n$-dimensional space, with $n$ indicating the number of variables involved in the optimization. The "black-box" nature of the problem makes solving such problems particularly challenging.

To address this black-box challenge, surrogate optimization algorithms Forrester & Keane (2009) model the target problem $f$ directly for guiding optimization. Among them, Bayesian Optimization

(BO) Shahriari et al. (2015) stands out as the most well-known method which leverages the uncertainty of Gaussian processes MacKay et al. (1998) to estimate the potential optimal locations of the problem. BO consists of three main components: first, a surrogate model, typically a Gaussian process, is constructed to predict the objective problem and quantify the uncertainty of the predictions; second, an acquisition function is used to select the input points most likely to lead to optimization based on the surrogate model's predictions; and finally, guided by the acquisition function, other optimization methods are employed to select candidate solutions, evaluate them, and then update the surrogate model. This process is repeated until end.

CIFAR

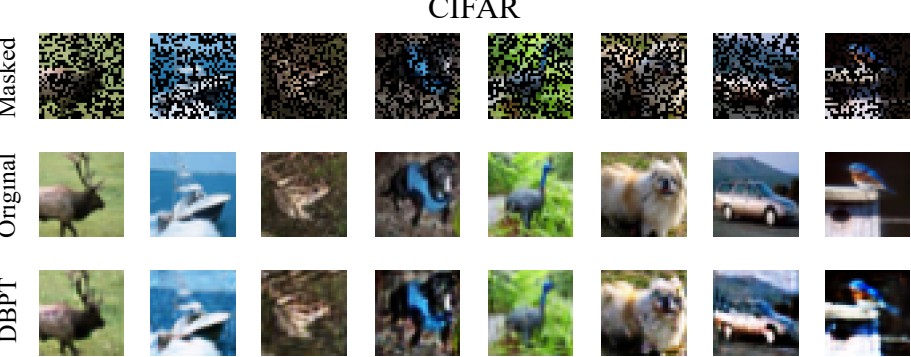

Figure 7: Image completion on CIFAR dataset. The "Masked" columns show the masked images, with unmasked pixels treated as known observations. The masked pixels are represented in black.

In this experiment, we use Gaussian Process-based Bayesian Optimization (GP-BO) as the baseline and replace the Gaussian process with WGP Wilson et al. (2011), CNP Garnelo et al. (2018a), markov Rabiner (2002), DKL Wilson et al. (2016), SDE Matching Bartosh et al. (2025), and DBPT to evaluate the performance of these algorithms from a black-box optimization perspective. The search domain is discretized into 200 candidate points. The initial point is drawn uniformly at random from these 200 candidates. At each iteration, we use Expected Improvement (EI) as the acquisition function and optimize EI by exhaustively evaluating it over all 200 candidates, selecting the maximizer. The evaluation budget is fixed at 35 objective evaluations. Because EI requires a predictive mean and variance at each candidate, for DBPT we obtain these moments by drawing 1000 independent process samples per iteration, computing the sample mean and variance at each of the 200 candidate points, and then plugging these estimates into the EI. We validate the results on two widely used multimodal black-box optimization problems, the Schwefel and Rastrigin problems Finck et al. (2010).

The mathematical expressions for these problems are as follows:

1. Schwefel problem:

$$f(\boldsymbol{x}) = 418.9829n - \sum_{i=1}^{n} x_i \sin(\sqrt{|x_i|})$$

2. Rastrigin problem:

$$f(\boldsymbol{x}) = 10n + \sum_{i=1}^{n} \left( x_i^2 - 10\cos(2\pi x_i) \right)$$

In the actual optimization process, we follow standard practices in the BBO field by intentionally ignoring any explicit expression information, treating the problem as a black-box, and optimizing without utilizing any inherent knowledge of the problem. For each experiment, we begin by randomly initializing three solutions, and in each iteration, we select the best candidate solution by optimizing EI.

## J    ABLATION EXPERIENCE

To evaluate the effectiveness of deconvolution in our process transformation framework, we conduct an ablation experience. We replace all deconvolution layers with MLPs and transformer architecture. Fig. 8 presents the results of ablation. The MLP- and Transformer-based variants fail to effectively capture inter-variable dependencies, producing predictive distributions that are irregular and scattered rather than smooth. In the few-shot regime, both models overfit the observed indices while collapsing toward near-zero predictions at unobserved locations—a "shortcut" that lowers the expected loss without recovering the true target law. As a result, uncertainty is miscalibrated and spatial/temporal coherence is degraded. In contrast, deconvolution, by combining upsampling and convolution operations, can effectively capture the spatial correlation between neighboring random variables Chen et al. (2022), thereby achieving better performance and smoother uncertainty estimates, especially in few-shot scenarios.

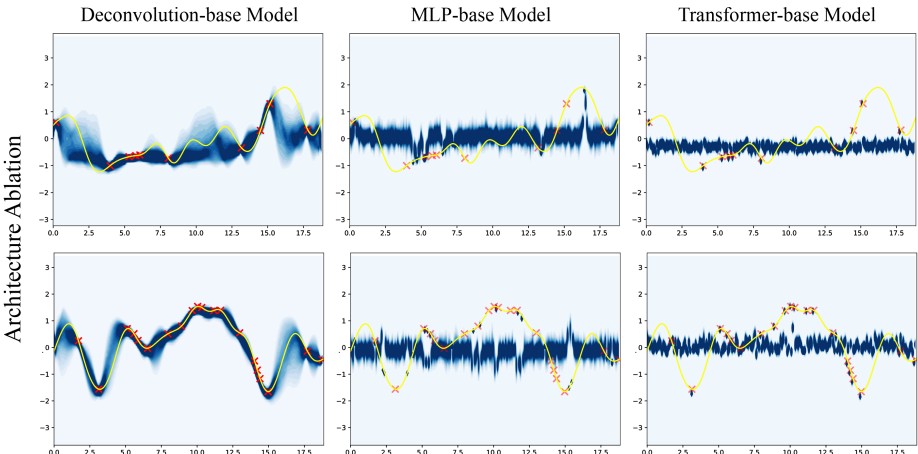

Figure 8: Ablation Experiment on deconvolution operation. Compared to the MLP and transformer architecture, the deconvolution operation effectively captures the correlations between neighboring random variables and enables smoother uncertainty estimation in few-shot setting.

## K    OUT OF DISTRIBUTION EXPERIMENT FOR CNP

Meta-learning methods typically require multi-trajectory data to capture the common features of the target problem. However, in certain scenarios, such as black-box (BBO) or expensive settings (high-fidelity CFD wing simulation), obtaining such multi-trajectory data is impractical. Although training can be conducted using randomly generated or existing multi-trajectory data, when the training data significantly deviates from the test data, it is prone to the Out-of-Distribution (OOD) challenge Sun et al. (2024). In this section, we conduct an OOD experiment for the meta-learning method Conditional Neural Processes (CNP) to assess the generalization ability of these methods when faced with multi-trajectory training data that deviates from the true problem distribution. The purpose of this experiment is to demonstrate that, when only single-trajectory data is available and the underlying distribution is unknown, forcing the generation of multi-trajectory data for training the model may be detrimental.

Specifically, we use all the digit 9 images from MNIST as the meta-dataset to train CNP, and during the testing phase, we evaluate the trained CNP using masked digit 7 images. Fig. 9 presents the final experimental results. Although digits 9 and 7 are somewhat similar, they have different underlying distributions, and when trained only on digit 9, the CNP incorrectly completes the masked image of digit 7 as digit 9. Even when we input all pixels of the digit 7 images into the trained CNP, the result remains the same. This phenomenon arises because the distribution of the original training data deviates from the target problem distribution, making it difficult for the algorithm to adapt to out-of-distribution data, leading to performance degradation and incorrect results Ye et al. (2022).

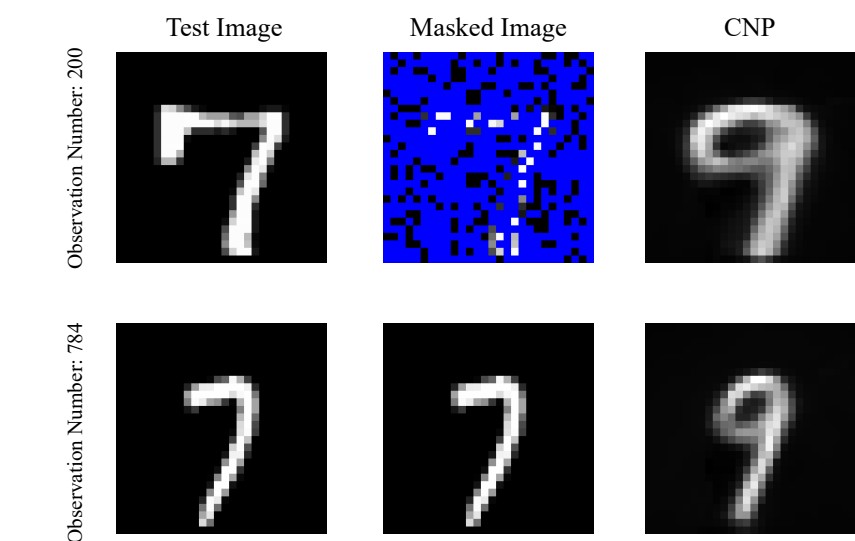

Figure 9: Out of Distribution Experiment for CNP. The masked pixels are represented in blue. The CNP trained on images of digit 9 incorrectly completes the masked image of digit 7 as digit 9.

## L    COMPUTER RESOURCES

The experiments are conducted on several machines, including machine A: Intel(R) Core(TM) i7-14650HX CPU, NVIDIA GeForce RTX 4060 Laptop GPU, running on Windows 11 operating system, and machine B: Dual Intel(R) Xeon(R) CPU E5-2620v3 2.40GHz CPU, NVIDIA Tesla A100 GPU, running on Linux operating system.

## M    COMPUTATIONAL COST

We benchmark the wall-clock training time of all methods on machine B for the synthetic task and image completion (MNIST, CIFAR), using each method's original architectural settings. Table 3 reports the results. GP and WGP exhibit increasing training time as the dataset size grows ($\approx$7/22 min and 8/24 min on MNIST/CIFAR, respectively), consistent with their higher computational complexity. DKL is comparatively costly, rising from ¡1 min on the synthetic task to 16 min on MNIST and 100 min on CIFAR. CNP completes training within a few minutes. SDE Matching requires 18 min on the synthetic task and was not evaluated on MNIST/CIFAR. In contrast, both the Markov baseline and DBPT complete training in under one minute across all settings. While DBPT's speed partly reflects its lightweight architecture, it nevertheless achieves strong performance, underscoring the advantages of the DBPT framework.

## N    THE USAGE OF LARGE LANGUAGE MODELS

A large language model (LLM) was used for manuscript polishing and to assist with intermediate steps in selected mathematical derivations; all results were verified by the authors.

## REFERENCES

Grigory Bartosh, Dmitry Vetrov, and Christian A Naesseth. Sde matching: Scalable and simulation-free training of latent stochastic differential equations. In *International Conference on Machine Learning (ICML 2025)*, 2025.

Table 3: Computational time of different algorithms. The unit is minutes. "$< 1$" indicates less than one minute, and "-" indicates that the data was not recorded.

| | SYNTHETIC TASK | MNIST | CIFAR |
|---|---|---|---|
| GP | $< 1$ | 7 | 22 |
| WGP | $< 1$ | 8 | 24 |
| MARKOV | $< 1$ | $< 1$ | $< 1$ |
| DKL | $< 1$ | 16 | 100 |
| SDE MATCHING | 18 | – | – |
| CNP | 2 | 2 | 7 |
| DBPT | $< 1$ | $< 1$ | $< 1$ |

Wessel P Bruinsma, Stratis Markou, James Requiema, Andrew YK Foong, Tom R Andersson, Anna Vaughan, Anthony Buonomo, J Scott Hosking, and Richard E Turner. Autoregressive conditional neural processes. *arXiv preprint arXiv:2303.14468*, 2023.

Bing Chen, Mazharul Islam, Jisuo Gao, and Lin Wang. Deconvolutional density network: Modeling free-form conditional distributions. In *Proceedings of the AAAI Conference on Artificial Intelligence*, volume 36, pp. 6183–6192, 2022.

Zexun Chen, Bo Wang, and Alexander N Gorban. Multivariate gaussian and student-t process regression for multi-output prediction. *Neural Computing and Applications*, 32:3005–3028, 2020.

Florinel-Alin Croitoru, Vlad Hondru, Radu Tudor Ionescu, and Mubarak Shah. Diffusion models in vision: A survey. *IEEE transactions on pattern analysis and machine intelligence*, 45(9): 10850–10869, 2023.

Andreas Damianou and Neil D Lawrence. Deep gaussian processes. In *Artificial intelligence and statistics*, pp. 207–215. PMLR, 2013.

James Durbin and Siem Jan Koopman. *Time series analysis by state space methods*. Oxford university press, 2012.

Vincent Dutordoir, Hugh Salimbeni, James Hensman, and Marc Deisenroth. Gaussian process conditional density estimation. In S. Bengio, H. Wallach, H. Larochelle, K. Grauman, N. Cesa-Bianchi, and R. Garnett (eds.), *Advances in Neural Information Processing Systems*, volume 31. Curran Associates, Inc., 2018. URL https://proceedings.neurips.cc/paper_files/paper/2018/file/6a61d423d02a1c56250dc23ae7ff12f3-Paper.pdf.

Steffen Finck, Nikolaus Hansen, Raymond Ros, and Anne Auger. Real-parameter black-box optimization benchmarking 2009: Presentation of the noiseless functions. Technical report, Citeseer, 2010.

Alexander IJ Forrester and Andy J Keane. Recent advances in surrogate-based optimization. *Progress in aerospace sciences*, 45(1-3):50–79, 2009.

Marta Garnelo, Dan Rosenbaum, Christopher Maddison, Tiago Ramalho, David Saxton, Murray Shanahan, Yee Whye Teh, Danilo Rezende, and S. M. Ali Eslami. Conditional neural processes. In Jennifer Dy and Andreas Krause (eds.), *Proceedings of the 35th International Conference on Machine Learning*, volume 80 of *Proceedings of Machine Learning Research*, pp. 1704–1713. PMLR, 10–15 Jul 2018a.

Marta Garnelo, Jonathan Schwarz, Dan Rosenbaum, Fabio Viola, Danilo J Rezende, SM Eslami, and Yee Whye Teh. Neural processes. *arXiv preprint arXiv:1807.01622*, 2018b.

Jonathan Gordon, Wessel P Bruinsma, Andrew YK Foong, James Requeima, Yann Dubois, and Richard E Turner. Convolutional conditional neural processes. *arXiv preprint arXiv:1910.13556*, 2019.

Daolang Huang, Manuel Haussmann, Ulpu Remes, ST John, Grégoire Clarté, Kevin Luck, Samuel Kaski, and Luigi Acerbi. Practical equivariances via relational conditional neural processes. In A. Oh, T. Naumann, A. Globerson, K. Saenko, M. Hardt, and S. Levine (eds.), *Advances in Neural Information Processing Systems*, volume 36, pp. 29201–29238. Curran Associates, Inc., 2023. URL https://proceedings.neurips.cc/paper_files/paper/2023/file/5d1a382162cb5ed326f1d3dbbfac4c82-Paper-Conference.pdf.

Miguel Lázaro-Gredilla. Bayesian warped gaussian processes. *Advances in Neural Information Processing Systems*, 25, 2012.

Minfang Lu, Shuai Ning, Shuangrong Liu, Fengyang Sun, Bo Zhang, Bo Yang, and Lin Wang. Opt-gan: A broad-spectrum global optimizer for black-box problems by learning distribution. In *Proceedings of the AAAI Conference on Artificial Intelligence*, volume 37, pp. 12462–12472, 2023.

David JC MacKay et al. Introduction to gaussian processes. *NATO ASI series F computer and systems sciences*, 168:133–166, 1998.

Joe Marino, Yisong Yue, and Stephan Mandt. Iterative amortized inference. In *International Conference on Machine Learning*, pp. 3403–3412. PMLR, 2018.

Juan Maroñas, Oliver Hamelijnck, Jeremias Knoblauch, and Theodoros Damoulas. Transforming gaussian processes with normalizing flows. In *International Conference on Artificial Intelligence and Statistics*, pp. 1081–1089. PMLR, 2021.

Parviz Mohammad Zadeh, Ali Mehmani, and Achille Messac. High fidelity multidisciplinary design optimization of a wing using the interaction of low and high fidelity models. *Optimization and Engineering*, 17(3):503–532, 2016.

Sebastian W. Ober, Carl Edward Rasmussen, and Mark van der Wilk. The promises and pitfalls of deep kernel learning. In *Proceedings of the 37th Conference on Uncertainty in Artificial Intelligence (UAI)*. PMLR, 2021. URL https://proceedings.mlr.press/v161/ober21a.html.

YongKyung Oh, Dongyoung Lim, and Sungil Kim. Stable neural stochastic differential equations in analyzing irregular time series data. In *International Conference on Learning Representations*, 2025.

Bernt Øksendal. Stochastic differential equations. In *Stochastic differential equations: an introduction with applications*, pp. 38–50. Springer, 2003.

Deep Shankar Pandey and Qi Yu. Evidential conditional neural processes. In *Proceedings of the AAAI Conference on Artificial Intelligence*, volume 37, pp. 9389–9397, 2023.

George Papamakarios, Eric Nalisnick, Danilo Jimenez Rezende, Shakir Mohamed, and Balaji Lakshminarayanan. Normalizing flows for probabilistic modeling and inference. *Journal of Machine Learning Research*, 22(57):1–64, 2021.

Lawrence R Rabiner. A tutorial on hidden markov models and selected applications in speech recognition. *Proceedings of the IEEE*, 77(2):257–286, 2002.

Sheldon M Ross. *Stochastic processes*. John Wiley & Sons, 1995.

Matthias Seeger. Gaussian processes for machine learning. *International journal of neural systems*, 14(02):69–106, 2004.

Marcin Sendera, Jacek Tabor, Aleksandra Nowak, Andrzej Bedychaj, Massimiliano Patacchiola, Tomasz Trzcinski, Przemysł aw Spurek, and Maciej Zieba. Non-gaussian gaussian processes for few-shot regression. In M. Ranzato, A. Beygelzimer, Y. Dauphin, P.S. Liang, and J. Wortman Vaughan (eds.), *Advances in Neural Information Processing Systems*, volume 34, pp. 10285–10298. Curran Associates, Inc., 2021. URL https://proceedings.neurips.cc/paper_files/paper/2021/file/54f3bc04830d762a3b56a789b6ff62df-Paper.pdf.

Amar Shah, Andrew Wilson, and Zoubin Ghahramani. Student-t processes as alternatives to gaussian processes. pp. 877–885. PMLR, 2014.

Bobak Shahriari, Kevin Swersky, Ziyu Wang, Ryan P Adams, and Nando De Freitas. Taking the human out of the loop: A review of bayesian optimization. *Proceedings of the IEEE*, 104(1): 148–175, 2015.

Mingchen Sun, Yingji Li, Ying Wang, and Xin Wang. Towards domain-aware stable meta learning for out-of-distribution generalization. *ACM Trans. Knowl. Discov. Data*, 18(8), August 2024. ISSN 1556-4681.

Belinda Tzen and Maxim Raginsky. Neural stochastic differential equations: Deep latent gaussian models in the diffusion limit. *arXiv preprint arXiv:1905.09883*, 2019.

Marc Vaisband, Valentin von Bornhaupt, Nina Schmid, Izdar Abulizi, and Jan Hasenauer. Loss formulations for assumption-free neural inference of sde coefficient functions. *npj Systems Biology and Applications*, 11(1):22, 2025.

Andrew Gordon Wilson, David A Knowles, and Zoubin Ghahramani. Gaussian process regression networks. *arXiv preprint arXiv:1110.4411*, 2011.

Andrew Gordon Wilson, Zhiting Hu, Ruslan Salakhutdinov, and Eric P Xing. Deep kernel learning. In *Artificial intelligence and statistics*, pp. 370–378. PMLR, 2016.

Nanyang Ye, Kaican Li, Haoyue Bai, Runpeng Yu, Lanqing Hong, Fengwei Zhou, Zhenguo Li, and Jun Zhu. Ood-bench: Quantifying and understanding two dimensions of out-of-distribution generalization. In *Proceedings of the IEEE/CVF Conference on Computer Vision and Pattern Recognition*, pp. 7947–7958, 2022.

Zesheng Ye and Lina Yao. Contrastive conditional neural processes. In *Proceedings of the IEEE/CVF Conference on Computer Vision and Pattern Recognition*, pp. 9687–9696, 2022.

Haibin Yu, Dapeng Liu, Bryan Kian Hsiang Low, and Patrick Jaillet. Convolutional normalizing flows for deep gaussian processes. In *2021 International Joint Conference on Neural Networks (IJCNN)*, pp. 1–6. IEEE, 2021.

