# OpenReview forum: "Noise-to-Process Transformation: A Weak-Prior Paradigm for Single-Trajectory Stochastic Process Modeling"
_ICLR.cc/2026/Conference — Submitted to ICLR 2026_

### Official Review · Reviewer_Rozw · 2025-10-20

**Soundness:** 3
**Presentation:** 2
**Contribution:** 2
**Rating:** 4
**Confidence:** 3

**Summary:**

This paper introduces the Noise-to-Process (N2P) framework, transforming a sample from a base-noise process (Z) into a single trajectory (X) that remains consistent with observed data, supported by substantial theoretical development. Building on this, the authors propose a Deconvolution-Based Process Transformation (DBPT) implementation, which shows improvements over existing baselines on the MNIST and CIFAR datasets.

**Strengths:**

* **Thorough literature review**: Clearly situates the paper within prior work and explains both its positioning and novelty.


* **Sound theoretical development**: Provides extensive theory development to justify the method design.

**Weaknesses:**

*  The biggest concern is that the empirical performance seems weak. The proposed method shows superior results on MNIST/CIFAR at (32x32), but performs worse than WGP on the BIA and PDB benchmarks. The baselines and datasets are also kind of weak: the paper should at least test at (64x64), and more challenging datasets would be welcome. Most baselines are pre-2018, with only one (Bartosh et al., 2025) that is recent, including more recent baselines will strengthen the experiment part.

* Line 96: the proof of Proposition 2 is referred to Proposition 10, but no proof is provided for Proposition 10.

* The discussion of “Prior-driven Approaches” is not accurate. Line 222 states, “Despite these advances, learning remains anchored to a predefined prior scaffold,” This is a strong claim, and the prior over stochastic processes can be learned directly from data via flow/diffusion models (e.g., Shi et al., 2025), which enables exact (or principled) posterior sampling.  Reference: Shi et al., Stochastic Process Learning via Operator Flow Matching, 2025.

* The paper introduces a theoretically index-agnostic paradigm but instantiates it with a specific, practical architecture (DBPT) that is, by its deconvolutional nature, tied to a discrete (regular) grid and its specific training resolution. The authors should explicitly acknowledge the gap between the theoretical advantages of the paradigm and the practical limitations of the implementation.

* Super-resolution experiments are missing. While the DBPT design seems applicable for super-resolution, its convolutional constraints likely limit evaluation to the specific training resolution. In contrast, general NP or operator-learning–based models often enable zero-shot evaluation at different resolutions (e.g., train on 64x64, evaluate on 128x128 or higher) without retraining.

*  The ablation study in Appendix J is confusing. The Transformer-based model seems to perform very poorly , while the deconvolution architecture is significantly better. This large performance gap raises suspicion that the Transformer model may not be correctly implemented or tuned. Given that numerous state-of-the-art models in computer vision and generative modeling use Transformers as backbones and consistently show advantages over convolution-based models, I strongly suggest the authors detail the settings for this part and consider trying either a standard ViT (Vision Transformer) or a (multi-layer) cross-attention architecture

* Typos :
1) Line 269. “Figure 2 present” should be “presents”
2) Line 292,  “GP demonstrate” should be “demonstrates”.
3) Line 302, “The synthetic experiment demonstrate” should be “experiments”
4) Line 1063, “rising from !1 min” check the typo

**Questions:**

See weaknesses. I’m inclined to raise my score if those concerns are resolved.

---

> ### Author Response · Authors · 2025-12-03
>
> Thank you for the clear and actionable feedback. We appreciate the reviewer’s emphasis on experimental rigor, scalability, and precise theoretical referencing.
>
> W1 (Overall strength of experiments; resolution and scalability; inclusion of BIA/PDB).
> We agree that pursuing SOTA performance is important, but a more comprehensive evaluation should also communicate a method’s limitations—i.e., what problems it is well-suited for and where it may underperform—so that future work can build on a realistic understanding of the approach. This is why we retained the BIA and PDB financial experiments, which highlight a challenging, high-volatility scenario for single-trajectory modeling.
> We also agree that evaluations on only 32×32 (CIFAR) / 28×28 (MNIST) are insufficient to fully support scalability claims. In the revision, we will add image completion and/or related generation experiments at ≥64×64 resolution (e.g., using more challenging datasets/settings), and report both performance and computational costs to better characterize scalability.
>
> W2 (Rigor: missing proof for Proposition 10).
> Thank you for catching this issue. We acknowledge the lack of a complete proof presentation for Proposition 10 in the current version. In the revised manuscript, we will provide the full proof (or a complete proof sketch with all necessary steps) and ensure that the cross-references between the main text and appendix are correct and unambiguous.
>
> W3 (Overly broad statements about prior-driven methods)
> We accept this criticism and will revise the wording to avoid overgeneralization. Our intended point is that many prior-driven approaches, even when learning parameters, still optimize within pre-specified structural families (e.g., kernel classes, state-space forms, or constrained dynamics parameterizations). We did not intend to claim that prior-driven methods cannot learn flexible priors from data. In the revision, we will add and discuss representative work leveraging flow/diffusion/operator-matching mechanisms to learn more expressive priors, and clarify how those approaches relate to and differ from our setting and goals.
>
> W4 (Deconvolution captures dependence; limitation on super-resolution / cross-resolution generalization)
> We agree with the concern. The deconvolution-based architecture provides a direct and effective inductive bias for modeling dependencies among random variables on a discrete grid. However, as currently instantiated, a model trained at a given resolution does not directly provide super-resolution capability without retraining on a higher-resolution grid. This choice allows the method to focus on—and perform reliably in—discrete-grid problems, which cover many practical scenarios (e.g., regularly sampled time series, pixel grids). In the revision, we will explicitly state this limitation and work to address it (e.g., via cross-resolution training/evaluation protocols and architectural modifications that enable better resolution transfer).
>
> W5  (Transformer ablation looks unexpectedly poor; need detailed settings and fair comparison)
> We understand the concern. In Appendix J, we observed that replacing the deconvolutional decoder with MLP/Transformer variants under the few-shot single-trajectory regime can lead to overfitting and “collapse/shortcut” behavior, degrading predictions on unobserved regions and miscalibrating uncertainty. To ensure a fair and transparent comparison, in the revision we consider trying either adcanced Transformer variants or across-attention architecture and we will provide full architectural and training details for the Transformer variants, including tokenization strategy, positional encoding, depth/width, optimization settings, and the hyperparameter search ranges.
>
> W6 (Typos/grammar and consistency checks)
> Thank you for pointing out these issues. We will correct the specific grammatical/typographical errors (e.g., “present(s)”, “demonstrates”) and perform a comprehensive

---

### Official Review · Reviewer_C7wr · 2025-10-22

**Soundness:** 2
**Presentation:** 2
**Contribution:** 2
**Rating:** 2
**Confidence:** 4

**Summary:**

The paper proposes a “Noise-to-Process (N2P)” framework, claiming to model stochastic processes from a single trajectory via a generative mapping (X = G_\theta(Z)) from a shared noise process. A specific instantiation, Deconvolution-Based Process Transformation (DBPT), is introduced and trained using a masked mean squared error (MSE) objective. The authors argue that this approach enables process-level uncertainty modeling under weak priors without requiring multiple trajectories, in contrast to Neural Processes (NP) or Gaussian Process (GP) models.

**Strengths:**

1.The paper is clearly written and attempts to unify process modeling ideas (GP, NP, neural SDEs) under a shared noise-to-function framework.

2.The experimental results cover diverse tasks (synthetic, time series, image completion) and demonstrate reasonable reconstruction quality.

**Weaknesses:**

The proposed N2P framework largely restates existing ideas found in Variational Implicit Processes, Normalizing Flow GPs, or Neural SDEs. The notion of generating a stochastic process via a measurable transformation of a base noise process (e.g., a Gaussian or Wiener process) is well established in prior literature. The contribution here is mainly terminological (“weak prior paradigm”) rather than methodological.

Despite the stochastic notation, the training objective reduces to a deterministic regression with noise regularization:
$$
L = E_Z[\frac{1}{\tau_o}|R_{\tau_o}(G_\theta(Z)) - O|_F^2].
$$
There is no explicit likelihood, no KL regularization, and no posterior inference—thus no genuine process-level probabilistic learning. In effect, the method behaves like a conditional generator (akin to a GAN without a discriminator) trained purely with an MSE loss.

The comparison with Neural Processes (NP) is also misleading. The claim that NP “requires multiple trajectories” is not accurate; NP frameworks can, in principle, operate on single trajectories, though with limited generalization. More importantly, NP remains a proper probabilistic model with explicit latent variables and variational inference, whereas N2P collapses to deterministic regression. The distinction the paper emphasizes (task-level z vs noise process Z) is not substantial.

The experimental evaluation is limited. Reported improvements over baselines are modest and could easily result from architectural capacity or convolutional inductive biases. There is no ablation study to isolate the effect of the proposed “noise process” component, and the claims of “single-trajectory learning” are not convincingly demonstrated—the model still relies on dense sampling along one trajectory, which effectively provides many supervision points.

The paper does not compare against recent and strong baselines in process learning and uncertainty-aware meta-learning, such as Attentive Neural Processes (Kim et al., 2019), Convolutional Conditional Neural Processes (Gordon et al., 2019), Transformer Neural Processes (Nguyen & Grover, 2022), and Neural Diffusion Processes (Dutordoir et al., 2023).

It also fails to cite or discuss several directly relevant works. Most notably, Variational Implicit Processes (Garnelo et al., ICML 2019) and “Functional Variational Inference based on Stochastic Process Generators” (Chao Ma, NeurIPS 2021) already introduced the same “noise-to-function” formulation with proper probabilistic objectives. Similarly, the idea of mapping base noise to structured samples has long existed in GANs and Normalizing Flow models. By omitting these foundational references and not clarifying its novelty relative to them, the submission overstates its originality and misrepresents its contribution.

The experimental validation is limited in both scale and diversity. Most experiments are confined to small, overused datasets such as MNIST and CIFAR-10. These benchmarks are no longer considered sufficient for demonstrating generalization or scalability in the ICLR community, as their challenges have been largely saturated. The paper does not evaluate on larger, more complex datasets or real-world continuous process data, making it difficult to assess whether the proposed framework meaningfully improves process-level modeling beyond toy examples.

While the paper’s narrative is appealing—“learning stochastic processes from a single trajectory under weak priors”—the technical substance does not support this claim. The approach amounts to deterministic regression with injected noise and lacks both probabilistic rigor and meaningful novelty. The comparison with Neural Processes is conceptually misleading, and the theoretical contributions are largely decorative.

Additionally, several presentation and technical issues reduce the clarity of the paper. Many equations appear without numbering, making it difficult to reference them in the text. In addition, some lemmas and corollaries are stated without proof or with only vague intuitive arguments. For a paper that emphasizes theoretical grounding, the absence of formal derivations undermines the claimed rigor and makes it hard to verify correctness.

**Questions:**

1. The paper claims to enable single-trajectory stochastic process learning, yet the training still relies on densely sampled points from the same trajectory. How does the method behave under sparse or partially observed data? Is there any theoretical or empirical analysis of sample complexity?

2. The notion of a “weak prior” is central to the paper’s narrative. Could the authors formally define what constitutes a “weak” prior in this context and explain how it differs quantitatively from priors in GP or Neural SDE models?

3. The training loss is a plain MSE （page 4） with Monte Carlo noise resampling. How does this loss capture process-level uncertainty rather than simple reconstruction accuracy? Have the authors considered using an explicit likelihood-based or probabilistic objective instead?

4. Compared with *Variational Implicit Processes* (VIP) (Garnelo et al., ICML 2019), the proposed method removes the variational posterior and KL term. What is the theoretical justification for this simplification? Does this mean the model is optimized purely under empirical risk minimization without probabilistic inference semantics?

5. Theoretical elements such as Kolmogorov consistency and measurability are presented at length. Do these properties impose any actual constraints or provide practical benefits for model training and inference, or are they purely formal?

6. The experiments omit comparisons with recent state-of-the-art Neural Process and process-learning models (e.g., Attentive NP, Transformer NP, Neural Diffusion Processes). Were these baselines tested, and if not, how do the authors justify the fairness of their empirical evaluation?

---

> ### Author Response · Authors · 2025-12-03
>
> Thank you for the detailed critique and the helpful pointers to related literature. We carefully revisited the manuscript in light of your comments and respond point-by-point below.
>
> W1. Similarity to flow-based GP / Neural SDE.
> We acknowledge the connection. Flow-GP/Neural-SDE methods typically follow a hybrid paradigm: a classical probabilistic scaffold (e.g., kernelized GP or SDE drift/diffusion form) augmented by neural networks for flexibility. In contrast, we propose an end-to-end neural framework for single-trajectory process learning that does not assume a specific GP/SDE prior family, but induces an implicit process distribution via a shared-noise, single-generator construction. We also agree the “noise-to-function” line is under-cited; in the revision we will add and discuss VIP and related work, clarifying both overlap (noise-driven function sampling) and differences (single-trajectory setting; intrinsic projective consistency from shared-noise joint generation).
>
> W2. “No explicit likelihood/KL
> A key clarification is that defining a coherent probabilistic process is distinct from training via explicit likelihood or VI: validity/coherence of the induced process distribution does not require optimizing ELBO/KL. That said, we agree the current training objective is limited: DBPT uses masked MSE, which neither directly maximizes full-data likelihood nor equals full Bayesian posterior inference. Our intended framing is: DBPT learns an implicit process family under sparse consistency constraints and architectural inductive bias, producing a sampleable predictive distribution via noise resampling; masked MSE is adopted mainly for stability in single-trajectory self-supervision. In revision, we will also report proper scoring rules (e.g., CRPS/energy score) and discuss the result.
>
> W3. Theoretical contribution.
> VIP performs explicit variational inference to approximate the posterior by optimizing an ELBO. In contrast, our framework is grounded in Kolmogorov consistency/existence: the shared-noise, single-generator pushforward construction induces projectively consistent finite-dimensional marginals, providing a principled basis that our model defines a coherent stochastic process over the considered index set, rather than relying on ELBO-based posterior inference. We will make this distinction clearer and expand the related-work discussion accordingly.
> Moreover, a key difference from many conventional conditional generative models is that they typically return only pointwise conditional densities at queried locations and do not explicitly represent joint, cross-index dependencies among random variables. Our approach enables direct modeling of inter-variable dependencies and yields subset predictions as projections of the same joint law. This distinction is already discussed in the main text.
>
> W4. Experiment results
> (1) Thank you for the comment. We will add ablations to isolate the effect of the noise-process/shared-noise component, and strengthen the “single-trajectory learning” claim by sweeping observation density (number of observed points / mask ratio) to verify performance under truly sparse supervision.
> (2) We agree the current evaluation is limited in scale and diversity. In the revision, we will include larger and more challenging datasets (including ≥64×64 image tasks)  and report scalability to better assess generalization beyond saturated benchmarks. We also agree current baselines  sufficient and will add stronger/more recent baselines to strengthen empirical evidence.
>
> Responses to Questions：
> Q1. Single-trajectory learning under severe sparsity
> We already tested an extreme sparse regime in synthetic experiments (only 10 observed points). Our method outperforms most baselines and approaches GP-level performance in both accuracy and uncertainty. In revision, we will add a theoretical complexity analysis to further substantiate this.
>
> Q2. Definition of “weak prior.”
> By “weak prior” we mean we do not assume a fixed kernel family (GP), fixed transition structure (Markov), or prescribed SDE drift/diffusion parameterization. Instead we impose minimal structure—shared i.i.d. noise + a single generator—inducing an implicit prior as a pushforward measure on path space, enabling diverse process forms without committing to a narrow parametric prior family.
>
> Q3. How can MSE capture uncertainty?
> Noise injection + masked MSE can capture uncertainty to some extent through stochastic sampling and is robust for single-trajectory self-supervision. Still, MSE is not a proper scoring rule; we will include CRPS/energy score and discuss benefits/feasibility (including potential replacement or mixing).
>
> Q4. Relation to VIP.
> Please see W3.
>
> Q5. Why is Kolmogorov theory relevant?
> It provides a principled basis: it guarantees the induced marginals are projectively consistent, supporting the claim that our construction defines a coherent stochastic process on the index set.
>
> Q6. Stronger baselines.
> Please see W4.

---

### Official Review · Reviewer_FTEz · 2025-10-27

**Soundness:** 2
**Presentation:** 2
**Contribution:** 2
**Rating:** 2
**Confidence:** 4

**Summary:**

This paper introduces a ``noise-to-process'' (N2P) paradigm for learning stochastic processes from a single, sparsely observed trajectory. The key idea is to learn a single, parameterized generator $G_{\theta}$  that maps a shared base-noise process $Z$ to a full trajectory $\bar{X} = G_{\theta}(Z)$. This design ensures projective consistency by construction, as all finite-dimensional marginals are projections of the same joint sample. The authors instantiate this paradigm with Deconvolution-Based Process Transformation (DBPT), which uses a noise encoder and a multi-scale deconvolutional decoder to capture inter-temporal dependencies. DBPT is evaluated on synthetic data, financial time series, image completion, and black-box optimization, comparing against prior-driven (e.g., GPs, SDEs) and data-driven (e.g., CNPs) baselines.

**Strengths:**

* The general idea of the paper is easy to follow.

* The `noise-to-process' paradigm, in its abstract form, bears a conceptual resemblance to frameworks like transformed GPs that map a base process through a nonlinear function. However, the significant novelty of this work lies in its concrete formulation for the single-trajectory regime and the introduction of the DBPT architecture, which uses a shared noise process and a deconvolutional decoder to explicitly enforce projective consistency and capture long-range dependencies.

**Weaknesses:**

Some limitations in my eyes are as follows:

* **Writting**. The writting and organization of the paper can be improved. For example, the citation commands (e.g., \citet vs. \citep) appear to be used wrongly and inconsistently, which affects the flow of the narrative. Several acronyms are introduced without full definitions at first use.

*  The experiment section can be better explained. For example, in terms of time-series MSE, what is the MSE here? Prediction or imputation?

*   **Limitation of Discrete Index Sets:** The entire framework is built upon a discretized index set $\mathcal{T}$. While Corollary 13 states that the model is *compatible* with Kolmogorov extension to a continuum, this is an existence result. In practice, the trained DBPT model is fixed to its training grid. Making predictions at arbitrary, new time points not in $\mathcal{T}$ would require re-discretization and potentially retraining, which is a significant limitation compared to native continuous-time models like GPs or Neural SDEs. The method lacks **native continuous-index inference**.

*   **Dependence on Generator and Noise Specifications:** The quality of the learned stochastic process is entirely dependent on the representational capacity of  $G_{\theta}$ and the characteristics of the base noise $Z $. While the deconvolution decoder is a good choice, the framework is susceptible to issues common in generative models, such as potential **mode collapse** or failure to capture the full complexity of the target process's randomness, especially if the architecture or noise dimension is poorly chosen.

*   **Scalability and Computational Cost:** The claim of ``lightweight computation'' (Appendix E) is supported for the presented tasks, but this may not scale well. Generating the *entire trajectory* in one forward pass means that for very high-resolution index sets (e.g., megapixel images or extremely long sequences), the memory and computation cost of the deconvolutional decoder could become prohibitive. A more nuanced discussion of the **computational complexity in  $|\mathcal{T}|$** and its scaling limits would be helpful.

*   **Depth of Comparison with State-of-the-Art:** The baseline selection is relatively dated (except SDE matching); a comparison with more recent and powerful sequence models, such as single-sequence diffusion models (see question section) or Gaussian process state-space models, would strengthen the evaluation.

**Questions:**

1. The theoretical guarantee of projective consistency is a key advantage. Could you design a simple quantitative experiment to empirically verify this property on a held-out test? For example, by showing that the marginal distribution at a point  $t$, computed from different higher-dimensional joint distributions that includet, remains consistent, which might not be the case for a method like CNP.

2. The masked MSE objective is simple, but it only supervises the mean (implicitly). While uncertainty emerges from noise resampling, the training signal doesn’t directly optimize calibration (e.g., via NLL or CRPS). Maybe it should note that this is a pragmatic choice, but may limit distributional fidelity compared to likelihood-based methods.  And also, in terms of performance metrics, including CRPS and other uncertainty quantification metrics would be beneficial.

3. In sparse data regimes, I suspect that overfitting can be an issue. At least, training the network here seems not data-efficient to me. Can the authors explain more about this, particularly compared to GP+DKL?

4. The paper shows an ablation on the decoder architecture. Could you provide more analysis on the sensitivity to the dimension and distribution of the base noise $Z$? What happens if  $d_z $ is too small or too large? Are there guidelines for choosing $Z$ for a new problem?

5. There are a series of papers about ``Diffusion Generative Models in Infinite Dimensions'' and ``Score-based Diffusion Models in Function Space'', which also transform the noise into a random process. There was a lack of discussion of comparisons in the paper. In my opinion, this paper should also compare to the Transformed Gaussian Processes (TGP) using Normalizing Flow, since TGP is also strongly related to this paper.

---

> ### Author Response · Authors · 2025-12-03
>
> Thank you for your thoughtful review and constructive suggestions. We appreciate your careful reading and your comments on both clarity and technical aspects. In the revision, we have corrected presentation issues (notation/abbreviations/citations), clarified the experimental protocol and evaluation metrics, and will further strengthen the empirical and theoretical support where you indicated gaps.
>
> W1–W2.  Thank you for the valuable feedback. We corrected several obvious errors (including abbreviation/notation issues), improved the overall narrative structure, and expanded the experimental descriptions to enhance readability.
> W3 (Discrete grids; theory vs. continuous indexing).  We agree that the current DBPT instantiation and experiments primarily target  discrete-grid  tasks (regularly sampled time series and image lattices), and our modeling assumption explicitly adopts a discrete index set (T). We emphasize two points:
> 1. Theory:  N2P's projective consistency and compatibility with Kolmogorov extension are not additional assumptions; they follow directly from the  pushforward construction  induced by “shared noise + a single generator.” For a nested grid family ({T_n}), the induced marginals are mutually consistent, hence compatible with Kolmogorov extension.
> 2. Instance / limitation. DBPT is not designed for strict interpolation or extrapolation at arbitrary off-grid continuous time points. Instead, it is intentionally tailored to discrete settings—such as regularly sampled financial time series and image lattices—where modeling trajectory-level dependencies on a predefined grid is the primary objective. We acknowledge that this pathwise formulation therefore lacks native continuous-index inference. In the revision, we will state this limitation explicitly and discuss plausible extensions, e.g., replacing the decoder with a coordinate-based implicit decoder that conditions on (t) to enable continuous indexing.
>
> W4 (Noise choice; mode collapse).
> 1. Noise distribution:  In theory, any i.i.d. base noise from a  non-atomic  distribution is measurably isomorphic on standard Borel spaces (“base-noise invariance”); therefore, the specific noise family is not a fundamental requirement of the framework.
> 2. Mode collapse:  DBPT does not use GAN-style adversarial training, and we did not observe collapse in our experiments. In contrast, our ablations indicate that replacing deconvolution with MLP/Transformer variants can cause  collapse on unobserved locations  (variance (\to 0)), leading to miscalibrated uncertainty.
>
> W5 (Scalability).  Our target setting is  single-trajectory, small-sample  modeling; many practical applications (e.g., stock time series) do not require million-scale resolutions. We will nevertheless add an explicit discussion of  time/memory complexity as a function of (|T|)  in the revision.
>
> W6 (Baselines).  We acknowledge the baseline suite is not strong enough. This is mainly because (i) work explicitly addressing  single-trajectory stochastic process  modeling is scarce, forcing reliance on older baselines, and (ii) many recent methods are designed for  multi-trajectory  regimes and can suffer severe generalization degradation without multi-trajectory data, making direct comparisons less meaningful. Still, we included a recent baseline (a Neural SDE method, ICLR 2025) in the original experiments. We agree this is insufficient, and in the revised version we have added several stronger baselines suggested by the reviewer to strengthen the empirical evidence.
>
> Responses to Questions“
> Q1. Standard CNP is largely  projectively consistent  with respect to the target set (T) (in the Kolmogorov sense). However, because CNP often assumes  conditional independence across target points  (lacking an explicit covariance structure), it remains limited in modeling  cross-target correlations , even if it satisfies target-set projective consistency.
> Q2. We agree. We currently train with  masked MSE , which is stable and suitable for single-trajectory self-supervision. We are considering adding  CRPS / energy score  and other sample-computable proper scoring rules as additional evaluation metrics, and discussing (or adding appendix experiments on) using them as training objectives or mixing them with MSE, including trade-offs (sample needs, gradient variance).
> Q3. Overfitting can occur. Our ablations show MLP/Transformer variants may “memorize” observed points and  collapse  on unobserved regions, causing uncertainty miscalibration; the deconvolutional structure substantially mitigates this issue.
> Q4. We will add new  noise-related ablations  (e.g., varying noise dimension) to quantify its impact. In addition, by base-noise invariance, the specific i.i.d. non-atomic noise family is not a fundamental limitation.
> Q5. Thank you for the pointer. We will include a more systematic discussion in  Related Work  and, where feasible, add stronger/more recent baselines to strengthen empirical evidence.

---

### Official Review · Reviewer_ZGqT · 2025-10-30

**Soundness:** 3
**Presentation:** 3
**Contribution:** 3
**Rating:** 2
**Confidence:** 2

**Summary:**

The work proposed a noise to stochastic process (N2P) framework that consists of two parts, a shared base process and a shared generator to transform the base process to observable trajectories. The work aims at solving the prior constraints of existing approaches like SDE-based approaches or structured GP-based approaches. The work further proposed an instantiation of the N2P framework called deconvolution based process transformation (DBPT) where the base process are IID Gaussians on a finite discrete time grid and the common generator is a deconvolution network. The work evaluate the proposed DBPT on both time-series modelling and image completion tasks and conducted ablation study on

**Strengths:**

1. The experiment section of the work considers a divers set of tasks including time series modeling, image modeling and black-box optimization.
2. The presentation is well structured with a general N2P framework followed by DBPT as the the concrete instantiation of the N2P framework and the technical details. The work makes a clear distinction between N2P as the theoretical framework and DBPT as a methodological contribution.
3. The work also studies different architecture choice for DBPT to justify the choice of a deconvolution architecture.

**Weaknesses:**

1. In Section 2.1, the work grounds the theoretical results on the basis of finite or countable time grid T. First, I can not see how a generator $G_\theta$ take an infinite number of $Z_t$s as inputs. Considering the actual DBPT model operates on a predefined, finite, discrete grid, the actual instantiation of the N2P framework is underwhelming and makes the grandiose theoretical result of Section 2.2 which invokes the Kolmogorov Extension Theory unnecessary.
2. The experiment results on finance related data is very weak.
3. The work compares against conditional neural processes and SDE matching as baselines from the SDE-based approaches and neural process families. More recent and stronger baselines like latent SDE [1, 2], attentive and transformer neural processes[3, 4], gaussian neural processes [5, 6] should be compared against.

References:

[1] Li, Xuechen, et al. "Scalable gradients for stochastic differential equations." International Conference on Artificial Intelligence and Statistics. PMLR, 2020.

[2] Deng, Ruizhi, et al. "Continuous latent process flows." Advances in Neural Information Processing Systems 34 (2021): 5162-5173

[3] Kim, Hyunjik, et al. "Attentive neural processes." arXiv preprint arXiv:1901.05761 (2019).

[4] Nguyen, Tung, and Aditya Grover. "Transformer neural processes: Uncertainty-aware meta learning via sequence modeling." arXiv preprint arXiv:2207.04179 (2022).

[5] Bruinsma, Wessel P., et al. "The Gaussian neural process." arXiv preprint arXiv:2101.03606 (2021).

[6] Markou, Stratis, et al. "Practical conditional neural processes via tractable dependent predictions." arXiv preprint arXiv:2203.08775 (2022).

**Questions:**

1. If we restrict the setup to a finite, discrete, and pre-defined time grid, is there any fundamental difference between the proposed N2P framework and the existing neural processes framework?

---

> ### Author Response · Authors · 2025-12-03
>
> We thank the reviewer for the careful evaluation and for highlighting key questions regarding the scope of the method, the discrete-grid instantiation, and the choice of baselines. We recognize that some parts of our exposition may have caused confusion, and we will revise the presentation to clearly the general N2P framework. We will also expand empirical comparisons and add clarifying discussions.
>
> W1(Finite vs. infinite-dimensional noise; role of Kolmogorov extension).
> We apologize for the unclear exposition that caused confusion. In our implementation, the generator G does not take an infinite-dimensional noise input. Instead, it takes a fixed-size noise tensor (matched to the predefined discrete grid) and outputs a discrete target trajectory. The model uses an encoder–decoder design, where the decoder is based on deconvolution/upsampling blocks, which provides an effective inductive bias for capturing local dependencies and correlations among random variables on the grid. While the current DBPT instantiation is restricted to finite, discrete, predefined grids, this restriction is intentional: under the single-trajectory and limited-observation regime, it yields a more stable learning procedure and a more reliable capture of local structure than many more flexible but less constrained alternatives.
> Regarding our theoretical discussion: we invoke Kolmogorov consistency/extension results to justify that the family of finite-dimensional marginals induced by a shared-noise, single-generator construction is projectively consistent. This theoretical perspective provides a principled foundation for the coherence of our process construction (even though training is performed on a discrete grid).
>
> W2 (Financial experiments appear weak).
> We agree that the financial time-series setting is particularly challenging: the data exhibit high volatility and complex nonstationarities, making it difficult to learn an accurate stochastic process from a single trajectory. We nevertheless included these experiments intentionally. While achieving SOTA performance is important, we believe a comprehensive evaluation should also reveal where a method is (and is not) effective. Reporting both strengths and limitations helps characterize the scope of applicability and may be beneficial for subsequent research building on or improving the framework. In the revision, we will further clarify this motivation and strengthen the empirical evidence (e.g., with additional metrics and analysis).
>
> W3 (Baselines are not strong enough).
> Thank you for pointing this out. We acknowledge that our baseline suite is not sufficiently strong. This is partly because single-trajectory stochastic process learning has relatively limited prior work, which restricts the availability of directly comparable baselines. In addition, many recent methods are primarily developed for multi-trajectory settings; when deprived of multi-trajectory data, their performance and generalization can degrade substantially, which motivated our conservative baseline selection. That said, we agree this is not fully satisfactory. We already included a recent baseline (a Neural SDE method) in our experiments, but we recognize this alone is insufficient. In the revised version, we will incorporate several of the stronger and more recent baselines you suggested (where applicable under the single-trajectory protocol), and we will report the comparison under a carefully controlled and fair evaluation setup.
>
> Q1 (If restricted to a finite, discrete, predefined grid, is N2P fundamentally different from Neural Processes?)
> We believe there remain three fundamental differences:
> 1.N2P directly defines and samples a full trajectory joint sample via a shared-noise single-generator construction. As a result, predictions on any subset are projections of the same joint sample/joint law, making projective consistency an intrinsic structural property. Standard NP/CNP formulations do not inherently enforce this unless additional architectural constraints are explicitly imposed.
> 2.N2P is explicitly formulated for single-trajectory learning under sparse observations. In contrast, Neural Processes are typically designed for learning function priors from multiple trajectories/tasks via amortized inference, which can be brittle when only a single trajectory is available.
> 3.One motivation of N2P is to generate an entire trajectory so as to directly model dependencies among neighboring indices. In our DBPT instantiation, convolution/deconvolution provides an inductive bias for such cross-index correlations. Many NP variants instead emphasize set-based conditioning and permutation invariance; cross-point dependence is often mediated indirectly (e.g., via global latents or attention), and under extreme single-trajectory sparsity this does not necessarily yield stable, trajectory-level coherent joint sampling.

---

### Meta-Review · Area_Chair_URGF · 2025-12-15

**Summary:**

The authors propose a method they coin noise2process, which is a measurable transformation of some base process defined by some shared noise distribution. The transformation is then instantiated via an encoder-decoder model and applied to a discretised space and Gaussian noise. The authors provide a diverse set of evaluations across synthetic data, time-series modelling, image completion, and black-box optimisation tasks.

**Reviewer Concerns:**

The main reviewers' concerns are (i) concerning novelty and how the work relates to the large body of work on similar constructions, (ii) weak baselines selected for the evaluation, (iii) a misleading general theoretical definition but a very specific and limited instantiation of the idea. After reading the paper, I tend to agree with those main concerns of the reviewers regarding the submission.

**Reviewer Scores:**

- (all reviewers): Given the rebuttal and limited presentation of additional experimental results, I do not expect that any of the reviewers are likely to increase their scores for this submission.

---

### Decision · Program_Chairs · 2026-01-26

Reject